# Synaptotagmin 7 is targeted to the axonal plasma membrane through γ-secretase processing to promote synaptic vesicle docking in mouse hippocampal neurons

**Jason D Vevea[1,2], Grant F Kusick[3,4], Kevin C Courtney[1,2], Erin Chen[3], Shigeki Watanabe[3,5], Edwin R Chapman[1,2]***

[1]Department of Neuroscience, University of Wisconsin-Madison, Madison, United States; [2]Howard Hughes Medical Institute, Madison, United States; [3]Department of Cell Biology, Johns Hopkins University, School of Medicine, Baltimore, United States; [4]Biochemistry, Cellular and Molecular Biology Graduate Program, Johns Hopkins University, School of Medicine, Baltimore, United States; [5]Solomon H. Snyder Department of Neuroscience, Johns Hopkins University, School of Medicine, Baltimore, United States

**Abstract** Synaptotagmin 7 (SYT7) has emerged as a key regulator of presynaptic function, but its localization and precise role in the synaptic vesicle cycle remain the subject of debate. Here, we used iGluSnFR to optically interrogate glutamate release, at the single-bouton level, in SYT7KO-dissociated mouse hippocampal neurons. We analyzed asynchronous release, paired-pulse facilitation, and synaptic vesicle replenishment and found that SYT7 contributes to each of these processes to different degrees. 'Zap-and-freeze' electron microscopy revealed that a loss of SYT7 diminishes docking of synaptic vesicles after a stimulus and inhibits the recovery of depleted synaptic vesicles after a stimulus train. SYT7 supports these functions from the axonal plasma membrane, where its localization and stability require both γ-secretase-mediated cleavage and palmitoylation. In summary, SYT7 is a peripheral membrane protein that controls multiple modes of synaptic vesicle (SV) exocytosis and plasticity, in part, through enhancing activity-dependent docking of SVs.

*For correspondence: chapman@wisc.edu

Competing interests: The authors declare that no competing interests exist.

## Introduction

Calcium affords remarkable control over myriad membrane trafficking events in cells. In presynaptic nerve terminals, $Ca^{2+}$ is particularly important as it regulates numerous aspects of the synaptic vesicle (SV) cycle, including modes of exocytosis, endocytosis, and several forms of synaptic plasticity. There are three modes of exocytosis: synchronous release, which occurs with a short delay following a stimulus, asynchronous release, which is characterized by a longer, variable delay following a stimulus, and spontaneous release, which occurs in the absence of electrical activity. The magnitude or rate of these modes can be influenced by previous synaptic activity to mediate various forms of short-term synaptic plasticity (*Barrett and Stevens, 1972*). Given the centrality of $Ca^{2+}$ in the SV cycle, considerable attention has been directed toward identifying the underlying $Ca^{2+}$ sensors that regulate this pathway (*Katz and Miledi, 1965*). The synaptotagmins (SYTs) are a family of proteins characterized by the presence of tandem C2 domains that often mediate binding to $Ca^{2+}$ and phospholipid bilayers (*Wolfes and Dean, 2020*). The most studied isoform is synaptotagmin 1 (SYT1),

which promotes rapid synchronous SV exocytosis (*Littleton et al., 1993*; *Geppert et al., 1994*) and clamps spontaneous release (*Littleton et al., 1993*; *Liu et al., 2014a*). SYT2 is a closely related isoform that is expressed in neurons in the cerebellum and spinal cord where it functions in the same manner as SYT1 (*Pang et al., 2006*). Other SYT isoforms are expressed throughout the brain and have distinct affinities for $Ca^{2+}$ and membranes. Some isoforms do not bind $Ca^{2+}$ at all while the others fall into three distinct kinetic groupings based on how fast they bind or unbind to membranes in response to changes in $[Ca^{2+}]$ (*Hui et al., 2005*).

Synaptotagmin 7 (SYT7) is a broadly expressed isoform (*Li et al., 1995*) implicated in aspects of SV release and at least two forms of synaptic plasticity (*Huson and Regehr, 2020*). Despite the growing understanding of its importance, the subcellular location of SYT7 remains the subject of vigorous debate. In PC12 cells, contradictory reports have localized SYT7 to the plasma membrane (PM) (*Sugita et al., 2001*), endo-lysosomal compartments (*Monterrat et al., 2007*), or dense core vesicles (DCVs) (*Wang et al., 2005*). Additionally, SYT7 was found on lysosomes in normal rat kidney (NRK) fibroblasts (*Martinez et al., 2000*), DCVs in chromaffin cells (*Fukuda et al., 2004*), and in nerve terminals from mouse hippocampus (*Jackman et al., 2016*). When taken together, there is general agreement that SYT7 resides in the secretory pathway and may be enriched on lysosomes, DCVs, or the PM, perhaps depending on the cell type. When SYT7KO mice were first generated, they showed a grossly normal brain structure and no observable neurological phenotype (*Chakrabarti et al., 2003*). However, inhibition of SYT7 through antibody blockade or recombinant fragment-mediated competition revealed defects in PM repair (*Reddy et al., 2001*), and the first SYT7 knockout (KO) studies found reduced rates of neurite outgrowth (*Arantes and Andrews, 2006*) and alterations in bone density homeostasis (*Zhao et al., 2008*), all stemming from deficiencies in lysosomal exocytosis. Additional studies revealed that changes in SYT7 expression alter DCV exocytosis in PC12 (*Wang et al., 2005*), adrenal chromaffin (*Schonn et al., 2008*; *Rao et al., 2014*), and pancreatic beta cells (*Gut et al., 2001*; *Li et al., 2007*; *Gauthier et al., 2008*; *Gustavsson et al., 2008*).

Early experiments, in which SYT7 was overexpressed (OE) in neurons, hinted at a role for SYT7 in the SV cycle by uncovering a complex endocytosis phenotype (*Virmani et al., 2003*). However, a subsequent electrophysiological examination of synaptic transmission concluded that there was no change in SV release or short-term synaptic plasticity in the SYT7KOs (*Maximov et al., 2008*). This was unexpected, because the high affinity of SYT7 for $Ca^{2+}$ and its slow intrinsic kinetics made this isoform a compelling candidate to serve as a $Ca^{2+}$ sensor for asynchronous release or for short-term plasticity (*Bhalla et al., 2005*; *Hui et al., 2005*). Consequently, in 2010, a role for SYT7 in asynchronous release during high-frequency stimulation (HFS) trains was described at the zebrafish neuromuscular junction (*Wen et al., 2010*) and then in hippocampal neurons from mice (*Bacaj et al., 2013*). Based on these studies, SYT7 appears to impact release only when more than one stimulus is given. Interestingly, SYT7 has been shown to promote asynchronous release from neurons after a single stimulus, but only after artificial ectopic expression of SNAP-23 (*Weber et al., 2014*). At the same time, SYT7 was found to mediate $Ca^{2+}$-dependent SV replenishment in response to HFS (*Liu et al., 2014b*). Two years later, *Jackman et al., 2016* demonstrated that SYT7 was required for paired-pulse facilitation (PPF), a form of plasticity in which release is enhanced in response to a second stimulus when applied shortly after a conditioning stimulus (*Regehr, 2012*). These authors also found that facilitation supported frequency-invariant transmission at Purkinje cell to deep cerebellar nuclei and at vestibular synapses in mice (*Turecek et al., 2017*). At granule cell synapses, they observed a role for SYT7 in facilitation and asynchronous release (*Turecek and Regehr, 2018*). Finally, a role for SYT7 in facilitation, asynchronous release, and SV replenishment was observed at GABAergic basket cell-Purkinje cell synapses (*Chen et al., 2017*).

Investigating the function of SYT7 during the SV cycle has proven to be a complex task. Initially found to have no influence on the SV cycle in KO studies, SYT7 has now been reported to fulfill several different functions at various types of synapses. To reconcile these phenotypes, and to gain insights into the underlying mechanisms, we examined SV exocytosis in wild-type (WT) and SYT7KO hippocampal synapses in dissociated cultures using an optical biosensor for glutamate (iGluSnFR) (*Marvin et al., 2018*). Moreover, to gain insights into the precise steps in the SV cycle that are regulated by SYT7, we carried out 'zap-and-freeze' (*Kusick et al., 2020*) electron microscopy (EM) experiments. Use of iGluSnFR allowed us to monitor glutamate release directly from single presynaptic nerve terminals, and 'zap-and-freeze' EM yielded novel insights into the membrane trafficking

events that occur within 5 ms of an action potential (AP). Furthermore, we examined the localization, post-translational modifications, and function of SYT7 in neurons using powerful new Janelia Fluor (JF) HaloTag ligands (HTLs) (*Grimm et al., 2017*) in conjunction with SYT7 retargeting strategies. We show that synapses lacking SYT7 exhibit subtle defects in asynchronous release, a complete disruption of PPF, and decreased rates of SV replenishment. We propose that these deficiencies originate, at least in part, from modest reductions in SV docking during activity. Surprisingly, we discovered that the amino-terminus of SYT7 is cleaved by the Alzheimer's disease-relevant γ-secretase complex; the stability and localization of SYT7 is dependent on this proteolytic processing step and concurrent palmitoylation. We propose that these modifications may be critical for the subsynaptic membrane trafficking of SYT7 and its role in supporting the SV cycle. Finally, by retargeting and restricting SYT7 to various membranes in the synapse, we show for the first time that SYT7 must localize to the PM to support asynchronous release, PPF, and SV replenishment.

## Results

### SYT7 influences presynaptic neurotransmitter release during short-term synaptic plasticity

To monitor SV exocytosis, we transduced the low-affinity (S72A) optical glutamate reporter iGluSnFR (*Marvin et al., 2018*) into cultured mouse hippocampal neurons. This allowed us to monitor glutamate release irrespective of confounding postsynaptic factors (*Wu et al., 2017*). We first used a single stimulus to analyze and compare the magnitude of glutamate release between WT and SYT7KO neurons, as well as the balance of synchronous and asynchronous release. Representative traces are shown in (*Figure 1a*), with peak $\Delta F/F_0$ quantitation in (*Figure 1b*); no significant differences in the magnitude of glutamate release between WT and SYT7KO neurons were observed. We used a 10 ms cutoff to distinguish between synchronous and asynchronous glutamate peaks, as described in earlier patch-clamp experiments (*Yoshihara and Littleton, 2002*; *Nishiki and Augustine, 2004*). We found a small (3% difference in medians or 1.8% according to the Hodges-Lehmann estimate), but statistically significant, decrease in asynchronous release from SYT7KO neurons in response to the single stimulus (*Figure 1c*). Previous comparisons examining release, triggered by a single AP, and monitored electrophysiologically, found no differences between WT and SYT7KO synapses (*Liu et al., 2014a*; *Chen et al., 2017*). The small change that we detected is likely due to the sensitivity afforded by using the iGluSnFR optical probe to directly monitor glutamate release, as compared to post-synaptic recordings.

Next, we examined PPF, a form of short-term synaptic plasticity. We note that the ratio of the first two responses is more generally termed the paired-pulse ratio (PPR). Here, we examined the PPF tuning window by interrogating glutamate release at 50-, 100-, 200-, and 500-ms interstimulus intervals. For WT synapses, we detected facilitation (~10%) using iGluSnFR at 50-ms interstimulus intervals, a mild decline at 100 ms, and a loss of PPF at 200- and 500-ms interstimulus intervals (*Figure 1d*). In SYT7KO neurons, PPF is absent (*Figure 1d*); hence this simplified system recapitulates the role of SYT7 in PPF that was reported using hippocampal slice preparations (*Jackman et al., 2016*). Quantifying the PPR, we found that SYT7KOs release approximately half the amount of glutamate in response to the second stimulus relative to the first stimulus at all intervals (*Figure 1e*). As emphasized above, no differences were observed when quantifying the magnitude of glutamate release triggered by the first stimulus between WT and SYT7KO neurons; again, differences emerged only after the second stimulus (*Figure 1—figure supplement 1a–b*). An advantage of the optical measurements utilized here is that they report the spatial distribution of transmission and can reveal the number of active synapses from one response to the next (synaptic recruitment). Interestingly, in WT neurons, the number of synapses that actively release glutamate in response to a conditioning pulse is maintained, while SYT7KO neurons deactivate ~10% of synapses following interstimulus intervals of 50, 100, and 200 ms. The PPR from WT and SYT7KO neurons became equal only at the 500-ms interstimulus interval (*Figure 1f*, *Figure 1—figure supplement 1c–f*). By visualizing 20 Hz PPF using a temporally color-coded maximum projection (*Figure 1—figure supplement 1g–h*), it is readily apparent that there is a near global decrease in the ability of SYT7KO synapses to release glutamate following a conditioning stimulus. Release triggered by the first stimulus is color-coded green and release from the second stimulus is color-coded magenta. Facilitation is visible as

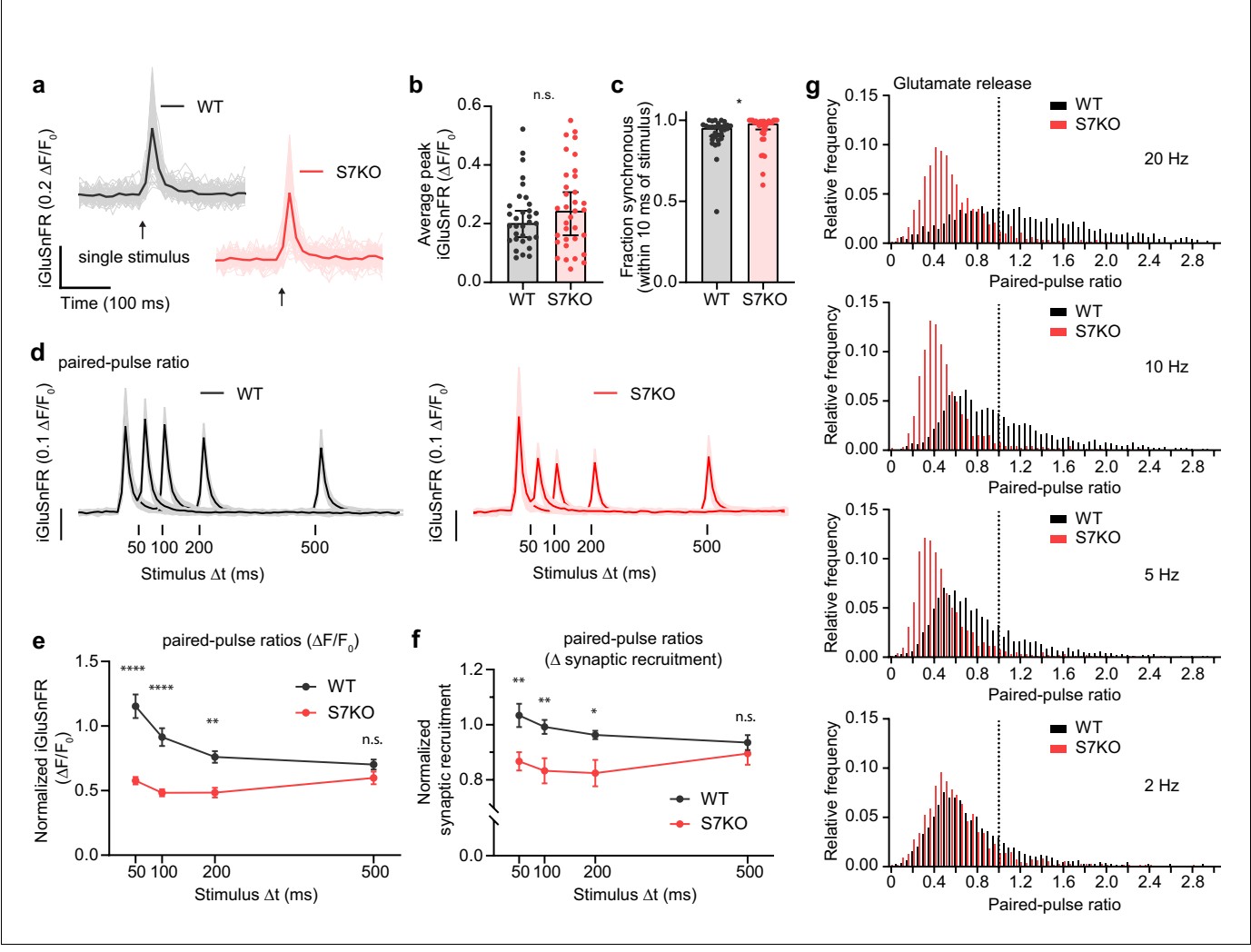

**Figure 1.** SYT7 influences presynaptic neurotransmitter release during short-term synaptic plasticity. (**a**) Representative super-folder iGluSnFR S72A (hereon iGluSnFR) traces from single-stimulus experiments. Lighter traces are individual regions of interest (ROIs) and dark bold traces are the average of all light traces from a full field of view (FOV); the single stimulus is denoted with an arrow. Wild-type (WT) are denoted in black and gray, and SYT7KO are represented in red and light red; same scheme applies throughout the figure. (**b**) Peak iGluSnFR signals between WT (0.203 [95% CI 0.154–0.244] $\Delta F/F_0$) and SYT7KO (0.245 [95% CI 0.160–0.308] $\Delta F/F_0$). Values are medians with 95% CI representing error, Mann-Whitney test, p = 0.4554, each n is a separate FOV (n = 32 (WT) and 34 (SYT7KO) from four independent experiments). (**c**) Fraction of synchronous release, defined as peak iGluSnFR signals arriving within 10 ms of stimulus from total release of 500 ms following the stimulus, compared between WT (0.9522 [95 % CI 0.902–0.965]) and SYT7KO (0.9808 [95% CI 0.943–0.993]). Data from the same n as in (**b**). Values are medians with 95% CI representing error, Mann-Whitney test, *p = 0.0326. (**d**) Average +/- standard deviation traces from paired-pulse ratio (PPR) experiments with four interstimulus intervals compared; n = 14 (WT 20 Hz), 14 (WT 10 Hz), 15 (WT 5 Hz), 13 (WT 2 Hz), 15 (SYT7KO 20 Hz), 13 (SYT7KO 10 Hz), 14 (SYT7KO 5 Hz), 13 (SYT7KO 2 Hz) from three independent experiments. (**e**) Quantification of PPR (peak iGluSnFR $\Delta F/F_0$) from WT and SYT7KO; values are means +/- SEM. ****p<0.0001, **p = 0.0012, by two-way analysis of variance (ANOVA) with Sidak's multiple comparisons test; full statistics are provided in *Figure 1—source data 1*. (**f**) Quantification of fractional active synapses, that is, the number of synapses demonstrating peak release above baseline during the second stimulus relative to the first of a paired pulse. Values are means +/- SEM. **p = 0.0052, **p = 0.0099, and *p = 0.0289, in order from left to right, by two-way ANOVA with Sidak's multiple comparisons test; full statistics are provided in *Figure 1—source data 2*. (**g**) Relative frequency histograms of PPR from all ROIs quantified from PPR trials, 20 Hz, 10 Hz, 5 Hz, 2 Hz, WT, and SYT7KO. Vertical dotted line delineates a PPR of 1.

The online version of this article includes the following source data and figure supplement(s) for figure 1:

**Source data 1.** Statistic summary using two-way ANOVA with Sidak's multiple comparisons test for quantification of PPR (peak iGluSnFR $\Delta F/F_0$) from WT and SYT7KO.

**Source data 2.** Statistic summary using two-way ANOVA with Sidak's multiple comparisons test for quantification of fractional active synapses, that is, the number of synapses demonstrating peak release above baseline during the second stimulus relative to the first of a paired pulse.

**Figure supplement 1.** Extended analysis of paired-pulse measurements.

*Figure 1 continued on next page*

Figure 1 continued

**Figure supplement 1—source data 1.** Statistic summary using two-way ANOVA with Sidak's multiple comparisons test for quantification of iGluSnFR $\Delta F/F_0$ peaks from second stimulation during PPR trials.

white or magenta while depression is visible as green. The relative frequency distributions of PPRs for 50-, 100-, 200-, and 500-ms interstimulus intervals are shown in (*Figure 1g*) where facilitating (PPR >1) and depressing (PPR <1) synapses are readily observable. These findings demonstrate that PPF is directly mediated by an enhancement of glutamate release from already active presynaptic boutons and not via recruitment of previously silent boutons. Hence, in WT synapses, SYT7 must somehow promote SV fusion during activity, perhaps by enhancing docking or stabilizing a docking intermediate; we address these possibilities further below.

## SYT7 counteracts synaptic depression and promotes asynchronous release during sustained stimulation

We further examined glutamate release from hippocampal synapses using iGluSnFR as described in *Figure 1*, but now as a function of HFS trains. The HFS consisted of 50 stimuli at 20 Hz (2.5 s stimulation epoch), which was sufficient to reach steady-state depression. Representative traces from individual regions of interest (ROIs) are shown in (*Figure 2a*) from WT (i) and SYT7KO (ii) neurons. Average iGluSnFR traces comparing WT and SYT7KO neurons during HFS show broad depression and a loss of tonic charge at SYT7KO synapses (*Figure 2b*). Similar findings were obtained via electrophysiological recordings of excitatory postsynaptic currents (EPSCs) (*Liu et al., 2014b*). During HFS, the average glutamate release declines in WT neurons, but this depression occurs more rapidly and deeply in SYT7KO neurons (*Figure 2—figure supplement 1a*; *Liu et al., 2014a*). Similarly, the number of active SYT7KO synapses that release glutamate also declined significantly (*Figure 2c*; *Figure 2—figure supplement 1b*). By measuring the cumulative iGluSnFR signal during an HFS, we calculated the SV replenishment rate (*Figure 2d*). This rate is the slope of a linear regression fitted to a steady state that is reached during the last 1.5 s of the HFS. SYT7KO synapses replenish SVs at about half the rate of WT synapses (*Figure 2e*), similar to what has been reported previously from electrophysiological measurements (*Liu et al., 2014a*; *Chen et al., 2017*). Interestingly, from the single ROI traces shown in *Figure 2a*, once release reached a steady state, fluorescent iGluSnFR responses decayed to baseline before the next stimulus. This contrasts with the average iGluSnFR fluorescence change, which does not decay to baseline (*Figure 2b*). We argue that the failure of the signal to decay to baseline in our average traces is analogous to the tonic charge component measured via electrophysiology and represents asynchronous release from single synapses. Therefore, we can use iGluSnFR imaging to monitor individual ROIs and measure HFS-related asynchronous release. Indeed, a smaller amount of asynchronous release is triggered during HFS at SYT7KO synapses relative to WT synapses, and this difference widens as stimulation progresses (*Figure 2f*). We emphasize that while we detected a small difference in asynchronous release between WT and SYT7KO neurons early in the train, this difference grew during the HFS train. Previous observations have relied on much stronger or longer stimuli, namely 10 s at 100 Hz (*Wen et al., 2010*) or 20 Hz (*Bacaj et al., 2013*), in order to detect the differences in train-related asynchronous release between WT and SYT7KO synapses. Our observations simplify the underlying mechanism by concluding that SYT7 is always acting to promote asynchronous release. This role is further enhanced during stimulus trains, perhaps owing to the slow intrinsic kinetics of SYT7 (*Hui et al., 2005*).

Next, we performed quantal analysis using the train stimulation data from WT neurons, comparing iGluSnFR signals from both early and late in the train. We analyzed release late in the train to define a single quantum, reasoning that uniquantal release predominantly occurs toward the end of a train when release reaches a steady state, and each synapse has a lower release probability. Binning peak iGluSnFR $\Delta F/F_0$ from early in the train (first two stimuli of 50 stimuli, 20 Hz train) results in a Gaussian distribution centered around a mean of 0.3 $\Delta F/F_0$ (*Figure 2g*), while peaks from the last five stimuli of the 20 Hz train resulted in a Gaussian centered around a mean of 0.14 $\Delta F/F_0$ (*Figure 2h*). Again, assuming single quanta are released late in a train at a steady state, this result supports the interpretation that early in a train, multiquantal release predominates. Indeed, a recent study demonstrated frequent multiquantal release from hippocampal neurons in response to single

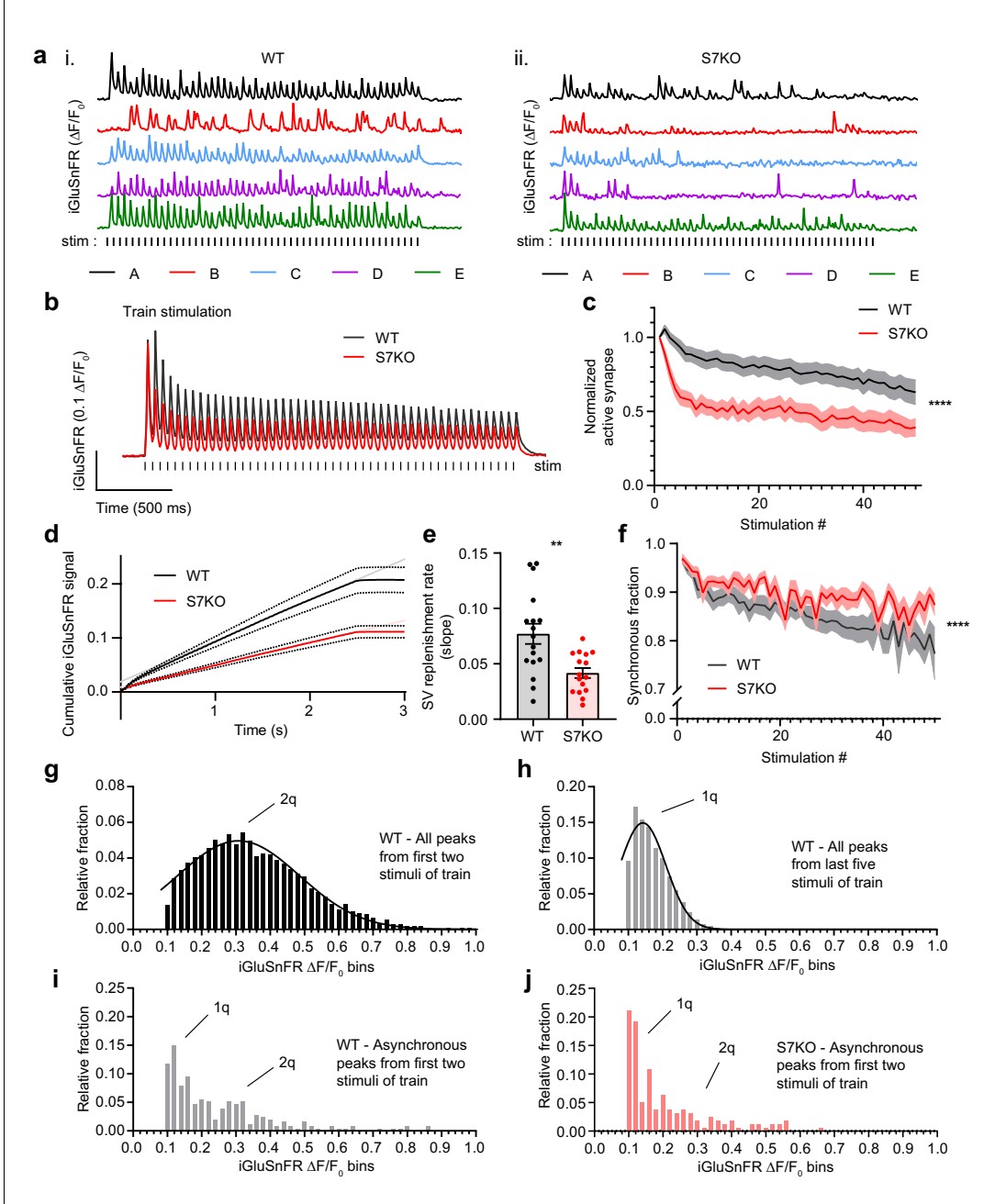

**Figure 2.** SYT7 counteracts depression and promotes asynchronous release during sustained stimulation. (**a**) Representative traces of iGluSnFR $\Delta F/F_0$ signals (single regions of interest (ROIs) A-E), from one full field of view (FOV) during high-frequency stimulation (HFS) of wild-type (WT) (i) and SYT7KO (ii) neuronal preparations. Samples were field stimulated with a frequency of 20 Hz for 2.5 s (50 action potentials (APs)). (**b**) Average iGluSnFR $\Delta F/F_0$ traces during high-frequency stimulation (HFS) for WT (black, n = 17) and SYT7KO (red, n = 16), from three independent experiments (same source data for **b**–**f**). (**c**) Fraction of active synapses, defined as synapses releasing peak glutamate above baseline, >4 SD above noise, as a function of stimulation number during HFS. Values are means (lines) +/- SEM (lighter shade error), ****$p<0.0001$ by two-way analysis of variance (ANOVA) comparing genotypes. (**d**) Plot of the average cumulative iGluSnFR $\Delta F/F_0$ signal from WT (black) and SYT7KO (red) neurons vs time. Dotted lines represent SEM and gray (WT) and light red (SYT7KO) linear lines represent linear fits to the last 1.5 s of the train. (**e**) Synaptic vesicle (SV) replenishment rates were calculated from slopes of linear regressions from individual traces used in panel (**d**). Values are means +/- SEM, WT (0.077 +/- 0.009) and SYT7KO (0.042 +/- 0.004); **$p = 0.0019$ using unpaired two-tailed t-test. (**f**) Fraction of synchronous release, defined as peak iGluSnFR $\Delta F/F_0$ within 10 ms of each stimulus from the total interstimulus interval, as a function of stimulation number during HFS. Values are means (bold lines) +/- SEM (lighter shade fill); ****$p<0.0001$ by two-way ANOVA comparing genotypes. (**g**) Quantal analysis using all detected iGluSnFR peaks (n>6000) from the first two stimuli of a 20 Hz train from WT neurons binned into 0.02 $\Delta F/F_0$. (**h**) Quantal analysis using all detected iGluSnFR peaks (n>10,000) from the last five stimuli of a 2.5-s 20 Hz train from WT neurons. (**i**) Quantal analysis using asynchronous iGluSnFR peaks (n = 254) from the first two stimuli of a train from WT

Figure 2 continued

neurons. (j) Quantal analysis using asynchronous iGluSnFR peaks (n = 156) from the first two stimuli of a train from S7KO neurons (asynchronous is defined as iGluSnFR peaks that occur more than 10 ms after a stimulus, but before the proceeding stimulus). Gaussian distributions were generated with no restrictions in panels (g) and (h). In panels (i) and (j), 1q and 2q labels were added based on the mean values from panels (g) and (h). From panel (g), mean (2q) = 0.31 [95% CI 0.30–0.32] and from panel (h), mean (1q) = 0.14 [95% CI 0.14–0.15]. WT asynchronous vs S7KO asynchronous distributions in panels (i) and (j) are different by Kolmogorov-Smirnov test; approximate p-value = 0.005 with K-S D = 0.1760.

The online version of this article includes the following figure supplement(s) for figure 2:

**Figure supplement 1.** Extended analyis of HFS experiments.

stimuli in WT neurons (**Kusick et al., 2020**). We interpret the rapid decline in release observed from SYT7KO neurons to reflect not only the rapid loss of uniquantal events (synaptic failures), but also the loss of multiquantal release events. Having defined the size of a quantum in our experiments, we applied these criteria to probe the nature of asynchronous release. We hypothesized that asynchronous release would comprise primarily single quanta throughout the train. However, we observed that in WT neurons, asynchronous release (events after the initial synchronous 10 ms window) was uni- and multiquantal, although uniquantal release was clearly favored (**Figure 2i**). Comparing WT and SYT7KO asynchronous release from the first two stimuli of a train, we observe a decreased fraction of multiquantal release events from SYT7KO neurons (**Figure 2j**). These data demonstrate a clear role for SYT7 in enhancing SV fusion during repetitive synaptic activity.

## SYT7 helps maintain docked and total synaptic vesicle pools after stimulation

To directly visualize the events that occur at synapses in response to single APs and HFS, and to understand how SYT7KO synapses depress faster, have less asynchronous release, and exhibit a much slower SV replenishment rate, we turned to 'zap-and-freeze' EM. This technique involves freezing synapses as fast as 5 ms after electrical stimulation, followed by freeze substitution and EM to observe synaptic ultrastructure. At rest, SYT7KO synapses have no gross morphological defects, with a normal complement of docked (in contact with the active zone PM) and total SVs in boutons (**Figure 3a–c**). In WT synapses, 40% of docked vesicles become undocked in response to a single AP, as previously reported (**Kusick et al., 2020**). Interestingly, 40% of vesicles are still undocked 5 ms after HFS, presumably because docked vesicle recovery matches depletion. By 5 s after HFS, the number of docked vesicles partly recover to baseline. A similar sequence of loss and recovery of docked vesicles was observed in SYT7KO synapses. However, in all conditions after stimulation, SYT7KO synapses had 30–40% fewer docked vesicles than the corresponding condition in WT (**Figure 3a–c**). It should be noted that this increased loss of docked vesicles is not due to increased depletion of vesicles by exocytosis, as indicated by iGluSnFR measurements above (**Figure 2**).

In response to a single stimulus, WT and SYT7KO neurons do not display any decreases in total SV number. However, following HFS, at the 5 ms time point, a modest decrease was observed in both conditions, and while WT synapses recovered 5 s after HFS, SYT7KO synapses did not (**Figure 3d**). Importantly, a careful analysis of the distribution of SVs within 100 nm of the active zone revealed that there were no changes, other than the docked pool, under any condition in WT or SYT7KO synapses (**Figure 3e**). This result demonstrates that the reduction in docking is specific and is not secondary to the reduction in the total number of vesicles near active zones.

The comparison of WT and SYT7KO synapses by 'zap-and-freeze' revealed two important observations that may help explain the complicated synaptic phenotype of the KOs. SYT7KO synapses display a greater loss of docked vesicles after a single stimulus and after HFS. Docking is a prerequisite to fusion; so decreases in docked vesicles after a stimulus could account for decreased asynchronous release, decreased PPF, and increased depression during HFS. Additionally, compared to WT synapses, SYT7KO synapses exhibit a decrease in the total number of SVs 5 s after HFS. This suggests that not only do SYT7KO synapses display a docking defect but they also suffer from an SV reformation defect lasting seconds after an HFS. SV docking and SV reformation are presumably two different processes and take place in different regions of the presynapse. To understand how SYT7 influences both processes, it is crucial to characterize the localization and trafficking of this protein.

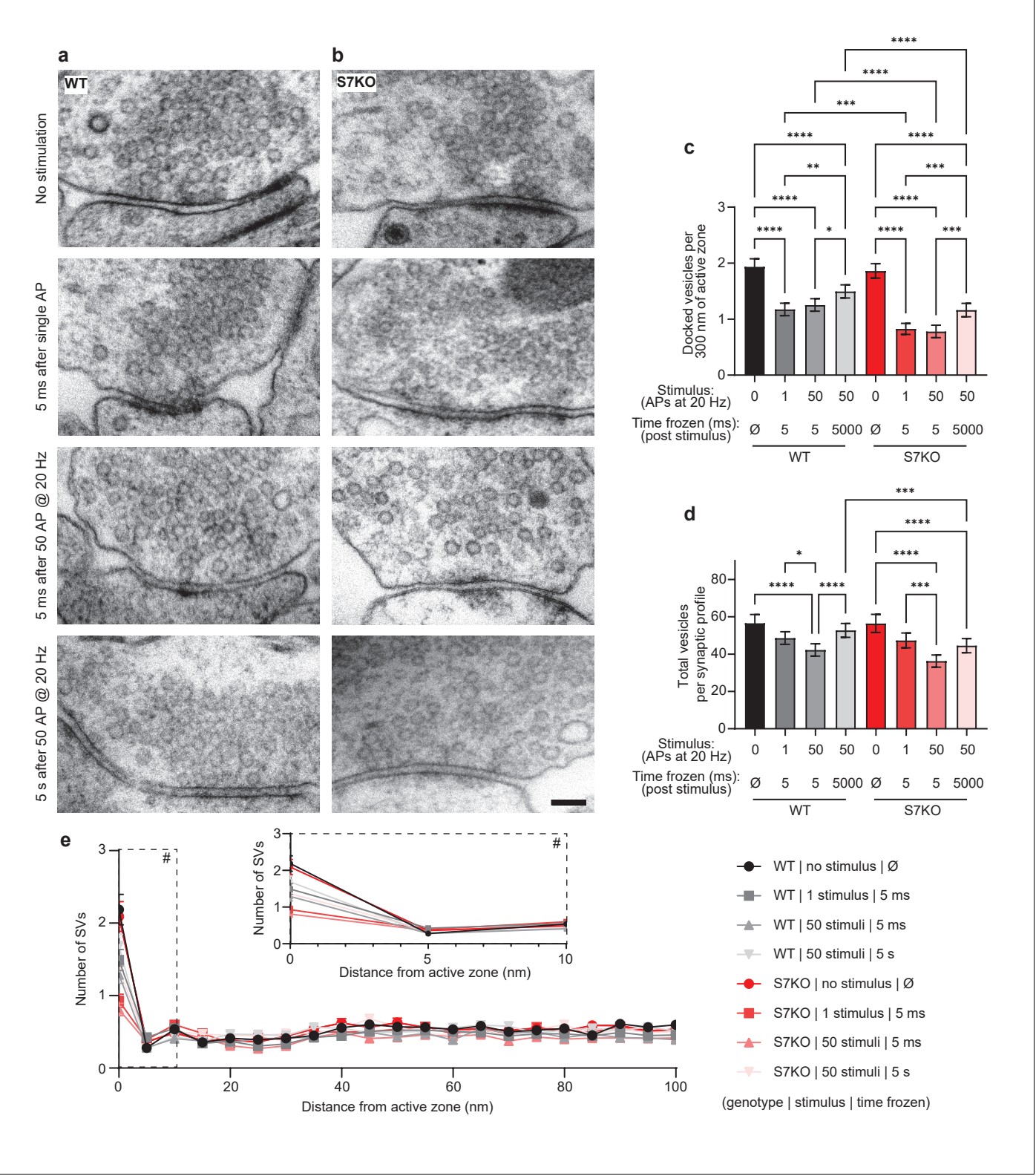

**Figure 3.** SYT7 enhances synaptic vesicle docking after stimulation. Representative electron micrographs of high-pressure frozen (a) wild-type (WT) and (b) SYT7KO synapses from labeled conditions. Scale bar = 100 nm. (c) Quantification of docked vesicle number normalized to 300 nm of active zone at rest, after stimulation with 1 action potential (AP) or 50 APs, and then frozen at 5 ms or 5 s post-stimulus. Docked vesicles are defined in high-pressure frozen samples as being in contact with the plasma membrane at the active zone (0 nm between the plasma membrane and vesicle membrane). WT conditions are in black to gray and SYT7KO conditions are in red to pink. Values are means +/- 95% CI and are from three biological replicates and

*Figure 3 continued on next page*

*Figure 3 continued*

over 300 n per condition (n = individual 2D electron microscopy (EM) images). All comparisons and summary statistics are provided in *Figure 3— source data 1*; ****p<0.0001, ***p<0.001, **p<0.01, and *p = 0.05, by Kruskal-Wallis test with Dunn's multiple comparison correction. (d) Quantification of vesicle number per 2D synaptic profile at rest, after stimulation with 1 AP or 50 APs, and then frozen at 5 ms post-stimulus or 5 s post-stimulus. Values are means +/- 95% CI and are from three biological replicates and over 2500 individual 2D EM images. All comparisons and summary statistics are provided in *Figure 3—source data 2*; ****p<0.0001, ***p<0.001, **p<0.01, and *p = 0.05, by Kruskal-Wallis test with Dunn's multiple comparison correction. (e) Quantification of synaptic vesicle (SV) number in relation to distance from active zone (az) up to 100 nm. Inset denoted with '#' sign is enlarged to show SV distribution in close proximity to az. Values are means +/- SEM.

The online version of this article includes the following source data for figure 3:

**Source data 1.** Statistic summary using Kruskal-Wallis test with Dunn's multiple comparison correction for quantification of docked vesicle number normalized to 300 nm of active zone.

**Source data 2.** Statistic summary using Kruskal-Wallis test with Dunn's multiple comparison correction for quantification of vesicle number per 2D synaptic profile.

## In hippocampal neurons, SYT7 is localized to both the axonal plasma membrane and LAMP1+ organelles that include active lysosomes

As outlined in the 'Introduction' section, SYT7 localizes to several distinct subcellular compartments in a variety of cell types; however, its localization in mature neurons remains unclear. In dissociated hippocampal neurons, endogenous untagged SYT7 is not detectable, in our hands, above background fluorescence by immunocytochemistry (ICC), presumably resulting from a mix of low expression and poor antibody performance. To localize SYT7α in mature neurons, we first sought to increase the expression of untagged SYT7α, via sparse lentiviral transduction (one transduction event per neuron), followed by detection using commercial antibodies and ICC. Using this approach, we found that SYT7α had a striking asymmetric localization to axons versus dendrites while also localizing to LAMP1+ structures in the soma (*Figure 4a*). Importantly, we used low levels of lentivirus so that we overexpressed just enough protein to detect with the antibody. We then expressed a SYT7α-HaloTag fusion protein, along with cytosolic mRuby3 (*Figure 4b*) or a PM-targeted msGFP (*Figure 4—figure supplement 1a*). SYT7α-HaloTag also exhibited a polarized distribution to axons (*Figure 4c*; *Figure 4—figure supplement 1b*), indicating that the carboxy-terminal HaloTag does not interfere with SYT7α trafficking. Axonal enrichment and dendritic exclusion suggest that the observed subcellular localization is not an overexpression artifact from the lysosomal compartment that resulted in nonspecific spill-over into the PM. Within axons, super-resolution Airyscan imaging further localized SYT7α-HaloTag to the PM (*Figure 4d–e*; *Figure 4—figure supplement 1c–d*). The line profile reveals cytosolic mRuby3 signal peaking in the middle of two characteristic 'double bump' signals from SYT7α-HaloTag, which resides on the plasma membrane. Indeed, mass spectrometry analysis of purified SVs fails to identify SYT7 as a principle component (*Takamori et al., 2006*), and SYT7 has been reported to be enriched in the synaptic PM in earlier fractionation experiments (*Sugita et al., 2001*).

To rule-out possible overexpression artifacts, we knocked-in a HaloTag at the endogenous locus of SYT7. For this, we constructed carboxy HaloTag homology-independent targeted integration (HITI) (*Suzuki et al., 2016*) vectors based on pORANGE cassettes (*Willems et al., 2020*). In our first attempt, single vector cloning was successful but sparse transfection failed to yield visible successful integrations. The SYT7 carboxy protospacer adjacent motif (PAM) sites have low scores for integration, and therefore, we split the pORANGE vector into two pFUGW-based lentiviral vectors (*Figure 4—figure supplement 1g*), one containing the spCas9 and the other containing the single guide RNA (sgRNA) and the HaloTag. Using these lentiviruses, and highly inclined and laminated optical sheet (HILO) microscopy (*Tokunaga et al., 2008*), we were able to detect fluorescent axons and soma compartments labeled with JF549 that were similar in pattern to our previous experiments overexpressing SYT7α (*Figure 4—figure supplement 1e*). Because the fluorescent signal was very weak, we attempted to use anti-HaloTag ICC for amplification. This was partially successful as axons now stained well and the asymmetric distribution of SYT7 to the axons versus dendrites was observed (*Figure 4—figure supplement 1f*); however, the anti-HaloTag antibody also stained the soma nonspecifically. Therefore, we continued to use slightly overexpressed SYT7α-

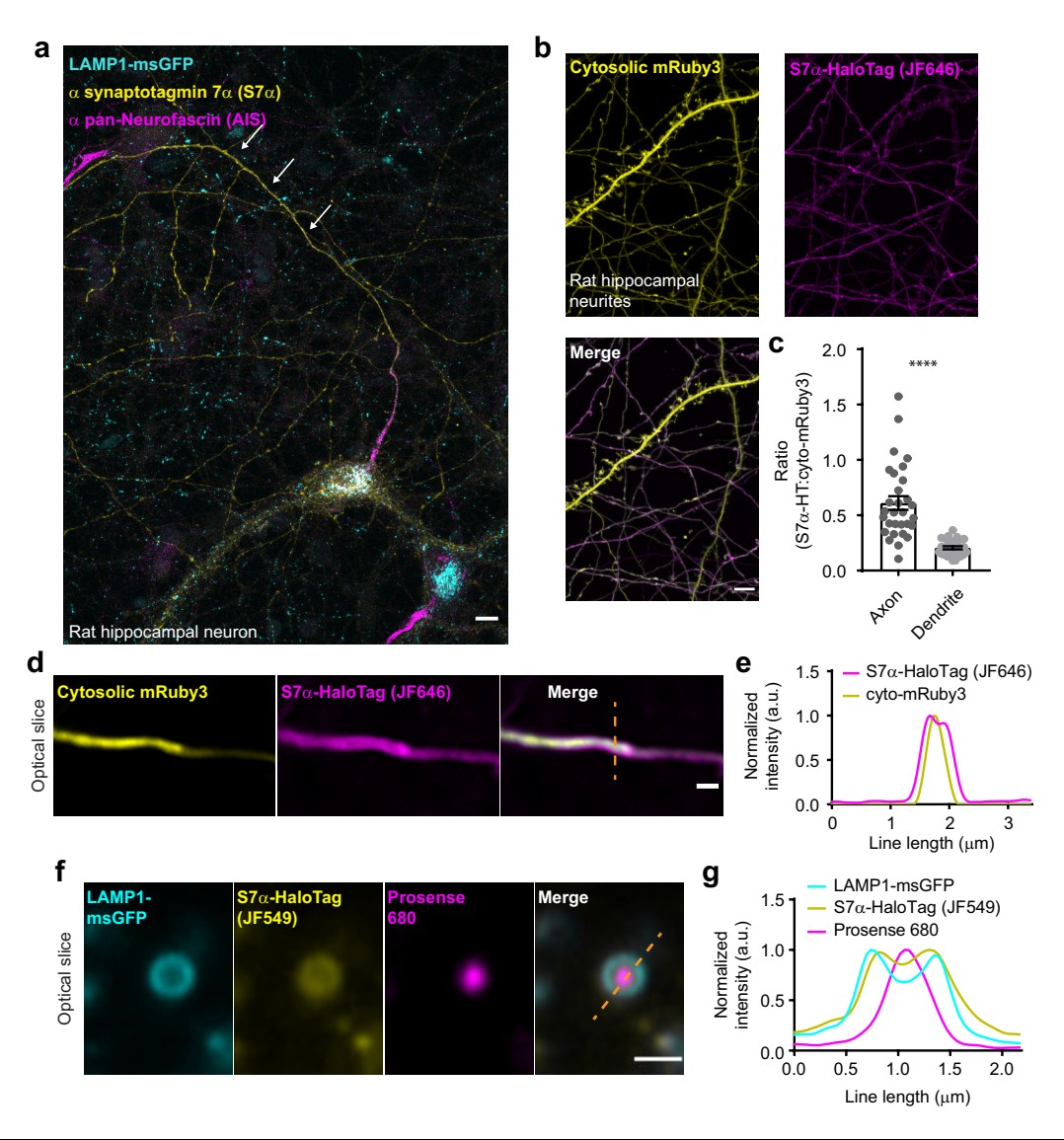

**Figure 4.** In hippocampal neurons, SYT7 is localized to both the axonal plasma membrane and LAMP1+ organelles that include active lysosomes. (**a**) Representative super-resolution fluorescent immunocytochemistry (ICC) image of rat hippocampal neurons at 15 days in vitro (DIV) expressing uniformly transduced LAMP1-msGFP and sparsely transduced, untagged SYT7α. These neurons were fixed and stained with antibodies to SYT7 (juxtamembrane region) and the axon initial segment (AIS) (anti pan-neurofascin). Scale bar = 5 μm. (**b**) Representative super-resolution images of cytosolically expressed mRuby3 (yellow/top left), SYT7α-HaloTag/JF646 (magenta/top right), and merged (bottom left). Scale bar = 5 μm. (**c**) Quantification of the ratio between fluorescent channels. Axonal ratio of SYT7-HaloTag:mRuby3 signal is 0.61 +/- 0.06, n = 30, while dendritic ratio is 0.21 +/- 0.01. Values are means +/- SEM from two independent experiments; p-value <0.0001 using unpaired two-tailed Welch's t-test. (**d**) Representative super-resolution optical slice of an axon (identified via morphology) expressing cytosolic mRuby3 (yellow) and SYT7α-HaloTag/JF646 (magenta). Merged image also denotes the line used in panel (**e**). Scale bar = 1 μm. (**e**) Plot of the normalized intensity profile along the orange dashed line in panel (**d**). (**f**) Representative super-resolution optical slice of a somatic lysosome from a rat hippocampal neuron at 16 DIV expressing LAMP1-msGFP (cyan), SYT7α-HaloTag/JF549 (yellow), and incubated with 0.5 μM Prosense 680 (magenta) for 12 hr. Scale bar = 1 μm. Merged image also denotes the line used in panel (**g**). (**g**) Plot of the normalized intensity profile along the dashed orange line in panel (**f**).

The online version of this article includes the following figure supplement(s) for figure 4:

**Figure supplement 1.** SYT7 is asymetrically localized to the axon.

**Figure supplement 2.** SYT7 is not a SV protein.

*Figure 4 continued on next page*

*Figure 4 continued*

**Figure supplement 3.** SYT7 does not influence steady-state cholesterol metabolism in neurons.

HaloTag, via sparse lentiviral transduction, to assess SYT7α localization as it was much brighter and localized to the same compartments as untagged SYT7α and endogenously (HITI) tagged SYT7.

Because SYT7α is localized to axons and influences the SV cycle, it was reasonable to predict that it might be translationally regulated, akin to bona fide SV proteins. The translation of SV proteins is correlated with synaptogenesis, so we probed synaptophysin (SYP), SYT1, SYT7, and total protein as a function of development and found that while SYP and SYT1 protein levels rise together, SYT7 does not follow the same trend as these SV proteins (*Figure 4—figure supplement 2a–b*). These observations provide further evidence that SYT7 does not localize to SVs but rather is targeted to another compartment.

To examine SYT7 localization in mature neurons with respect to the endo-lysosomal system, we co-expressed SYT7α-HaloTag, LAMP1-msGFP, and cytosolic mRuby3. Indeed, we observed broad colocalization of SYT7α-HaloTag and LAMP1-msGFP in the soma (*Figure 4—figure supplement 2c–e*). To further localize SYT7α in the soma, we counterstained SYT7α-HaloTag with antibodies against secretory pathway markers, including endoplasmic reticulum (ER), cis-Golgi, trans-Golgi, post-Golgi vesicles, and endosomes. We again found that SYT7α-HaloTag was highly colocalized with LAMP1-msGFP and, to a lesser extent, to the trans-Golgi and post-Golgi vesicles that were marked by sortilin (*Figure 4—figure supplement 2f–g*).

LAMP1-msGFP identifies mature lysosomes as well as intermediates in the endo-lysosomal compartment (*Cheng et al., 2018*). To specifically identify active lysosomes, we incubated neurons with Prosense 680. This molecule is self-quenching and membrane impermeant; when cleaved by lysosomal proteases it dequenches, and thus fluorescently labels active lysosomes (*Weissleder et al., 1999*). Interestingly, SYT7α-HaloTag was present throughout the endo-lysosomal compartments, on active and inactive lysosomes (*Figure 4f–g*). Importantly, SYT7α-HaloTag is clearly limited to the lysosomal membrane and does not appear to simply colocalize with lysosomes via a degradation pathway.

SYT7 has been localized to lysosomes in non-neuronal cells where it was reported to play a role in lysosomal exocytosis (*Martinez et al., 2000*). It is also reported to play a role in trafficking cholesterol by regulating lysosome-peroxisome interactions (*Chu et al., 2015*). Cholesterol is a lipid that is critically important for the formation of SVs; cholesterol also binds and regulates interactions between some SV proteins (*Thiele et al., 2000*). This link between SYT7 function, cholesterol trafficking, and the SV cycle is attractive because it might explain some of the SV cycle-related phenotypes of SYT7 deficient synapses. We therefore investigated cholesterol levels and interactions that are sensitive to changes in the abundance of this lipid, in SYT7KO neurons. More specifically, the SV proteins SYP and synaptobrevin (SYB) interact in a cholesterol-dependent manner (*Mitter et al., 2003*). If a loss of SYT7 results in decreased trafficking of cholesterol to the PM, as reported in HEK293T and SV589 cells (*Chu et al., 2015*), we should observe decreased cholesterol-dependent protein-protein interactions. Using mature neurons and a chemical crosslinker previously shown to successfully probe SYP/SYB interactions (*Mitter et al., 2003*), we did not observe decreased SYP/SYB interactions in SYT7KO neurons relative to WT (*Figure 4—figure supplement 3a–c*). Similarly, we did not see a change in any lipid species by thin layer chromatography (*Figure 4—figure supplement 3d–e*) or a buildup of neutral lipids in lysosomes (*Figure 4—figure supplement 3f*), as would be expected from a cholesterol trafficking defect. Based on these data, we conclude that SYT7 likely influences the SV cycle from its location on the axonal PM and not indirectly by altering the abundance or distribution of cholesterol in neurons. How SYT7 becomes enriched in axons and how it persists on the axonal PM despite robust membrane cycling during exo- and endocytosis, are questions that we explore in the next series of experiments.

## SYT7 is cleaved by the intramembrane aspartyl protease presenilin

In our efforts to localize SYT7α, we transduced neurons with a variety of tags at its amino- and carboxy-termini. When examining the expression levels of these constructs by immunoblot analysis, we observed that constructs tagged at their amino-termini existed as a mix of proteins with the

predicted (large) molecular weight of the fusion protein along with bands of apparently the same molecular weight as the untagged protein. In contrast, constructs tagged at their carboxy-termini yielded a single band that corresponded to the size of the full-length protein plus the tag (*Figure 5—figure supplement 1a–b*). Therefore, the artificial N-terminal tag is cleaved off by a cellular protease, and this cleavage must occur near the tag junction, or within the amino-terminus of SYT7α. Changing the tag or linker, or deleting the luminal domain, did not affect cleavage of SYT7α (data not shown), thus leaving the transmembrane domain (TMD) as the only possible cleavage site. Cytosolic-side cleavage is unlikely as there are palmitoylation sites on that side of the TMD that influence localization in fibroblasts (*Flannery et al., 2010*).

There are only a limited number of intramembrane proteases in cells. Interestingly, with their short luminal tail segments, SYTs have been postulated to be targets of the γ-secretase complex (*Südhof, 2002*); we tested this idea using inhibitors. Remarkably, a combination of 1 µM DAPT (N-[N-(3,5-difluorophenacetyl)-L-alanyl]-S-phenylglycine t-butyl ester) (a competitive presenilin inhibitor) and 20 µM GI 254023X (an ADAM10 metalloprotease inhibitor) strongly inhibited proteolytic processing of the HaloTag-SYT7α construct (*Figure 5—figure supplement 1c*). DAPT alone appeared to only prevent processing of an already cleaved form of HaloTag-SYT7α; GI 254023X had to be present as well to prevent the cleavage of HaloTag-SYT7α protein. However, when we transduced neurons with untagged SYT7α, only DAPT shifted the mobility of SYT7α when subject to sodium dodecyl sulfate-polyacrylamide gel electrophoresis (SDS-PAGE) (*Figure 5—figure supplement 1d*). These findings indicate that there are two cleavage reactions, one pertaining to the artificial tag, and another targeting the untagged SYT7α protein. Endogenous SYT7 is alternatively spliced to create at least three different isoforms with varying juxtamembrane linker lengths (*Fukuda et al., 2002*). We found that DAPT, but not GI 254023X, shifted the apparent size of all native SYT7 isoforms from rat hippocampal neurons (*Figure 5a*). This suggests that SYT7 isoforms do not need to be preprocessed by a metalloprotease and are bona fide direct targets of the γ-secretase complex.

The $IC_{50}$ for DAPT inhibition of γ-secretase-mediated cleavage of endogenous SYT7 was ~71 nM (*Figure 5b–c*), which is similar to the $IC_{50}$ of another γ-secretase target, amyloid precursor protein (APP) (*Dovey et al., 2001*). To address the location of SYT7 processing by γ-secretase, we transduced neurons with HaloTag-SYT7α and added an impermeant nonfluorogenic JF dye (JF549i) (*Xie et al., 2017*) to the media at a low concentration (1 nM) for 2 days. HaloTag-SYT7α that transits through the PM before cleavage (processing by γ-secretase) will become labeled with extracellular JF549i, allowing us to follow the intracellular fluorescent adduct. However, if SYT7 is processed at or before it reaches the PM, fluorescence will not be observed (*Figure 5d*). Indeed, full-length SYT7α is present on the PM and is apparently cleaved in synaptic endosomal structures because we observed small JF549i punctae (yellow) throughout the soma that partially colocalize with LAMP1 (cyan)-positive structures (*Figure 5e*). Interestingly, these punctae localize to the lumen of LAMP1-positive structures (i) or to a portion of the endo-lysosomal membrane (ii), or do not colocalize with LAMP1 at all (iii) (*Figure 5f*). As a control, JF549i did not label untransduced neurons, so all labeling was specific for tagged SYT7α (*Figure 5—figure supplement 1e*). These experiments revealed that SYT7 is first trafficked from the secretory pathway to the axonal PM where it is then subsequently processed by γ-secretase in an intracellular compartment.

## SYT7 is mislocalized and destabilized when amino-terminal cleavage is blocked

It remained unclear whether γ-secretase processing supports the axonal localization or function of SYT7 or whether this processing step is part of the normal degradation pathway for this protein. Interestingly, SYT7 is palmitoylated near its TMD, and this post-translational modification has been shown to be important for its trafficking in fibroblasts (*Flannery et al., 2010*). Here we examined SYT7 localization in control neurons and neurons treated with DAPT, 2-bromopalmitate (2-BP, a palmitoylation inhibitor), and DAPT + 2-BP (*Webb et al., 2000*). Proteins with palmitoylation sites are dynamically de-palmitoylated and re-palmitoylated; adding a palmitoylation inhibitor biases the protein to a de-palmitoylated state. For these experiments, we included SYT1 as a control. SYT1 is responsible for fast, synchronous SV fusion, and like SYT7, it is palmitoylated in or near its lone TMD (*Chapman et al., 1996*; *Chapman, 2008*). We also included LAMP1-msGFP as another general membrane-anchored protein control; this construct also allowed us to examine the colocalization of SYT7α and LAMP1+ structures.

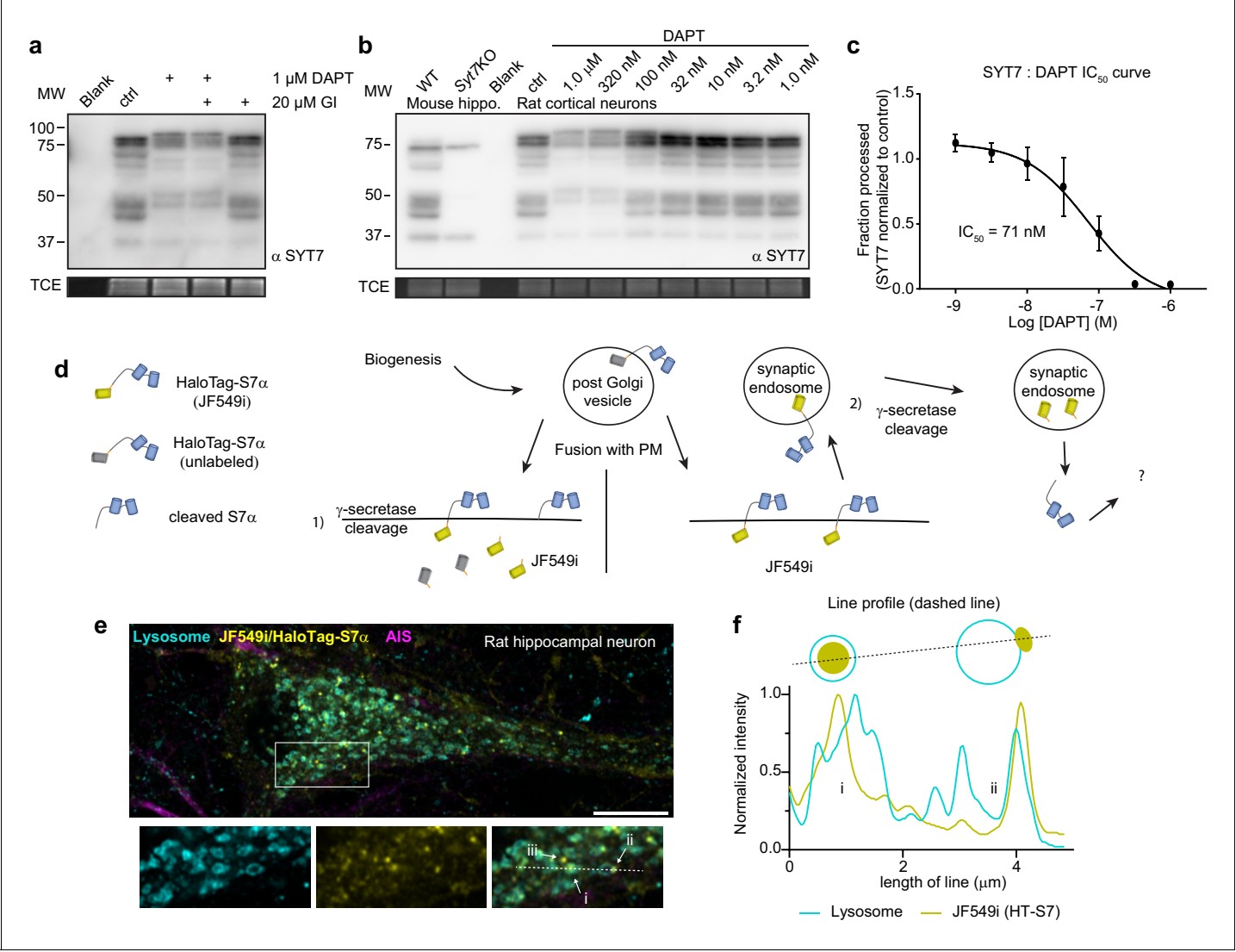

**Figure 5.** Synaptotagmin 7 is cleaved by the intramembrane aspartyl protease presenilin. (**a**) Representative anti-SYT7 immunoblot from rat hippocampal neurons with trichloroethanol (TCE) staining as a loading control. Conditions from left to right are blank/no protein, control conditions, neurons ttreated with 1 μM DAPT (N-[N-(3,5-difluorophenacetyl)-L-alanyl]-S-phenylglycine t-butyl ester) (presenilin competitive inhibitor), DAPT and 20 μM GI 254023X (ADAM10 selective inhibitor), or treated with GI 254023X only, all from DIV 5 onward. (**b**) Representative anti-SYT7 immunoblot using mouse hippocampal neurons for wild-type (WT) and SYT7KO antibody controls along with rat cortical neurons grown in various concentrations of DAPT to assay half maximal inhibitory concentration (IC$_{50}$), with TCE staining as a loading control. (**c**) Graph of the fraction of processed synaptotagmin 7 (SYT7) when grown in various DAPT concentrations in relation to control conditions (IC$_{50}$ curve) results in an IC$_{50}$ of 71 nM. The lowest specific SYT7 band was used for quantitating cleavage and IC$_{50}$ of DAPT. Values are means +/- SD after log transformation from three independent experiments. (**d**) Cartoon illustrating the logic and methodological approach to determine whether full-length SYT7 protein transits through the plasma membrane (PM) prior to amino-terminal cleavage by γ-secretase. JF549i is a membrane-impermeant version of JF549 (JF549 and JF549i are nonfluorogenic). In (1), cleavage can take place in the post-Golgi vesicle, prior to axonal PM localization or cleavage happens at the PM. No fluorescent HaloTag is observable in this scenario. In (2), SYT7 transits through the PM before being cleaved in a synaptic endosome. Only in this scenario will fluorescent HaloTag be observable in neurons. (**e**) Representative super-resolution optical slice of a rat hippocampal neuron transduced with LAMP1-msGFP (cyan) and HaloTag-SYT7α (yellow). Before fixing neurons, they were incubated with 1 nM HTL-JF549i for 2 days. Fixed neurons were decorated with anti-pan-neurofascin (magenta) antibodies to mark the axon initial segment (AIS). White box indicates the area that is enlarged to show the detail below the image. The labels (i), (ii), and (iii) indicate areas where HTL-JF549i appears inside lysosomes, clustered on the edge of lysosomes, or completely independent of lysosomes, respectively. (**f**) Line profile from the dashed line in panel (**e**) with normalized intensity of LAMP1-msGFP (cyan) and JF549i (yellow). The labels (i) and (ii) are labeled on the line profile as well and correspond to the same labels as in panel (**e**). Cartoon schematic of the analyzed signal is above the graph.

The online version of this article includes the following figure supplement(s) for figure 5:

**Figure supplement 1.** SYT7 is cleaved at its amino-terminus.

Neurons were untreated (control), treated with DAPT for 10–12 days, with 2-BP for 3 hr, or both, and then fixed and stained for SYT1, SYT7, and the axon initial segment (AIS). None of these treatments affected the localization of SYT1 (*Figure 6a*), whereas DAPT treatment resulted in the mislocalization of the majority of SYT7α-HaloTag to small punctae with only faint axonal staining. Surprisingly, a brief treatment with 2-BP led to the complete disappearance of SYT7α-HaloTag, as did the combination of DAPT and 2-BP (*Figure 6b*). The punctate SYT7α-HaloTag-positive structures observed during DAPT treatment appeared at the detriment of normal axonal and lysosomal localization (*Figure 6c–d*); under these conditions, SYT7α-HaloTag mislocalizes to the earlier secretory pathway at the expense of the later secretory pathway, as observed by the change in PCC between the two conditions (*Figure 6d*). These data support the hypothesis that γ-secretase is needed for SYT7 localization. Thus, treatment of WT neurons with DAPT could potentially phenocopy SYT7KO neurons, but this was not the case in our model system (*Figure 6—figure supplement 1a–c*). This lack of an effect might arise from low residual levels of axonal PM-targeted SYT7 that linger during DAPT treatment. Nevertheless, SYT7 is mislocalized upon γ-secretase inhibition and is also reliant on palmitoylation for stability and localization.

SYT7 has been reported to be a long-lived presynaptic protein, so we next investigated whether γ-secretase processing influences its half-life (*Dörrbaum et al., 2018*). We confirmed that SYT7α is indeed long-lived and found that γ-secretase inhibition enhances its turnover (*Figure 6e–f*; *Figure 6—figure supplement 1d*). This was somewhat unexpected because γ-secretase processing is conventionally thought to accelerate the turnover of its substrates (*Kopan and Ilagan, 2004*). Additionally, substitution of the palmitoylated cysteine residues with alanines results in an unstable protein when expressed in neurons, which is only marginally stabilized upon DAPT treatment (*Figure 6—figure supplement 1e*). These experiments revealed that γ-secretase processing and palmitoylation both play an essential role in determining SYT7 stability. In summary, SYT7 is cleaved in its TMD by γ-secretase, making it completely reliant on palmitoylation to associate with the PM.

## Dissociating discrete SYT7 functions via protein retargeting

We demonstrated above that SYT7 influences PPF, asynchronous release, and SV replenishment, and its cellular location and stability are regulated by γ-secretase processing and palmitoylation. We therefore asked whether the location of SYT7 influences these modes of release and found that the distinct functions of SYT7 in the SV cycle could be dissociated by retargeting the protein to different destinations. For these experiments, we restricted SYT7α to the PM, endo-lysosomal LAMP1+ membranes, or SVs, by replacing the luminal amino-terminus and TMD from SYT7α with different targeting motifs. To target SYT7α to the PM, we added a binding immunoglobulin protein (BiP) leader sequence followed by a CD4 TMD and a Golgi export sequence (*Figure 7a*; *Figure 7—figure supplement 1a*). For endo-lysosomal membrane targeting, fusing the cytosolic portion of SYT7α to the carboxy-terminus of LAMP1 was sufficient (*Figure 7b*). Using lentivirus, we did not detect LAMP1-SYT7α-HaloTag on the PM but we could detect it on the axonal PM upon overexpression using lipofectamine (*Figure 7—figure supplement 1b*). Therefore, the potential for spillover to the PM should be considered when interpreting results from this construct. Similarly, for targeting to SVs, we fused the cytosolic domain of SYT7α to SYP (*Figure 7c*; *Figure 7—figure supplement 1c*). Note, in *Figure 7a–b* and *Figure 7—figure supplement 1c*, retargeted SYT7α constructs were sparsely transduced to clearly demonstrate cellular localization. When expressed in HEK293T cells, these constructs also localized to the PM, endo-lysosomal compartment, and small vesicles, respectively (*Figure 7—figure supplement 1d*). Interestingly, SYT7α that was restricted to the PM by replacing the WT TMD with a CD4 TMD, and adding a viral Golgi export sequence, demonstrated a polarized distribution to axons. Therefore, while γ-secretase processing is an essential prerequisite for enrichment of WT SYT7α to the axonal PM, there is another axonal targeting motif in the protein (*Figure 7a*; *Figure 7—figure supplement 1a*).

To examine the function of these rescue constructs, we chose to use HFS so that we could measure (1) 20 Hz PPF, (2) train-related asynchronous release, and (3) synaptic depression and SV replenishment. For these experiments, we used a new floxed SYT7 mouse line (MRC Harwell Institute #Syt7-TM1C-EM4-B6N). This inducible KO avoids any developmental confounding factors due to chronic loss of SYT7 and serves to reduce animal waste. These experiments also validate our SYT7KO phenotypes in a separate genetic line and establish this new SYT7 floxed line for future use (*Figure 7—figure supplement 1e–f*). The expression of all rescue constructs was confirmed via

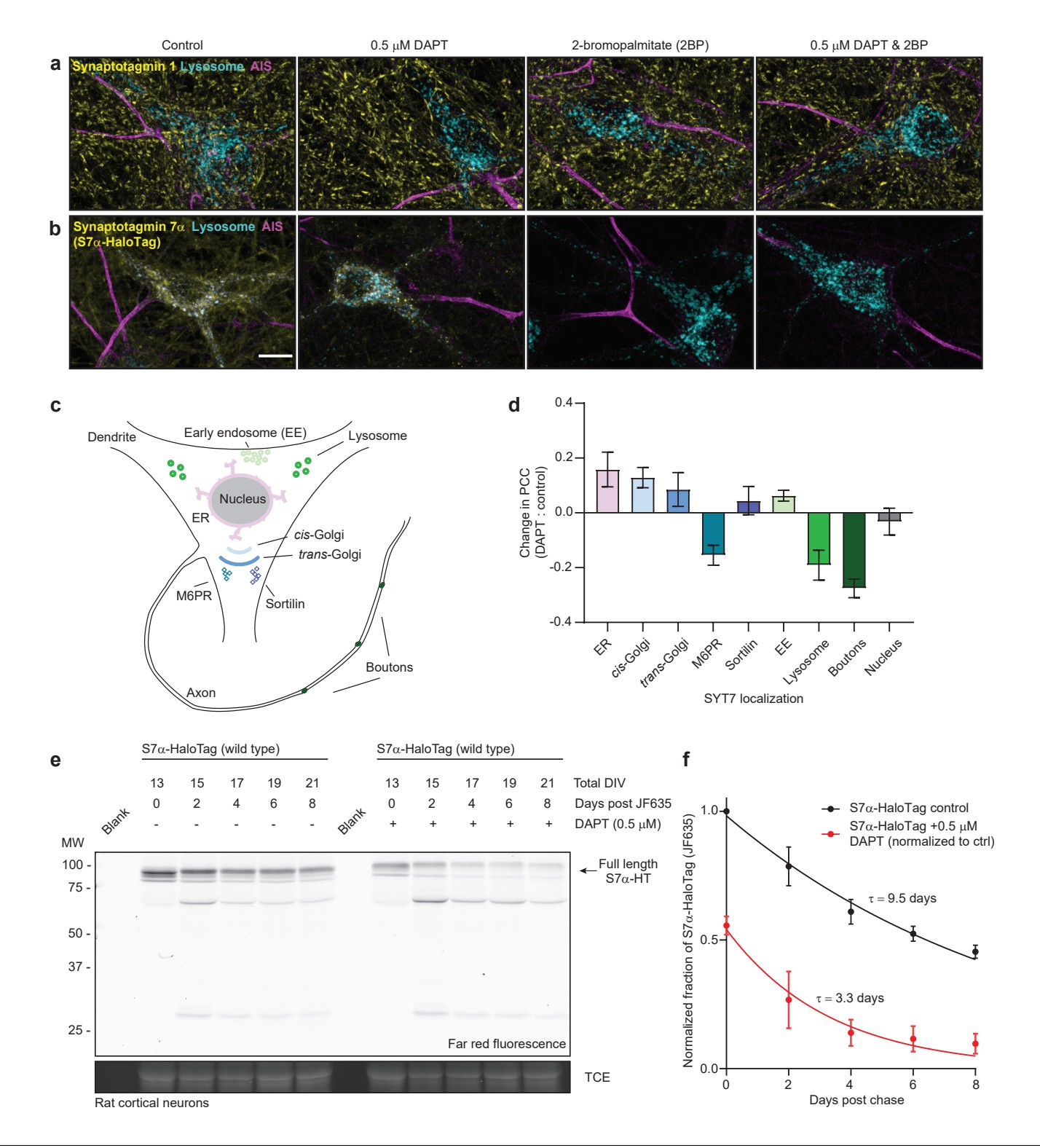

**Figure 6.** SYT7 is mislocalized and destabilized when amino-terminal cleavage is blocked. (**a**) Representative super-resolution maximum z-projections of rat hippocampal neurons transduced with LAMP1-msGFP (cyan), fixed for immunocytochemistry (ICC), and stained for synaptotagmin 1 (SYT1) (yellow) and the axon initial segment (AIS) (magenta). Four separate conditions were imaged: control neurons, neurons grown for 10–12 days in 0.5 μM DAPT (N-[N-(3,5-difluorophenacetyl)-L-alanyl]-S-phenylglycine t-butyl ester), neurons exposed to 2-bromopalmitate (2-BP) for 3 hr before imaging, and neurons exposed to a combination treatment of DAPT and 2-BP. (**b**) Same as in panel (**a**), but instead of anti-SYT1 staining, neurons were transduced

*Figure 6 continued on next page*

*Figure 6 continued*

with SYT7α-HaloTag and reacted with JF549 during overnight primary antibody incubation to monitor SYT7α localization. Scale bar = 10 μm. (**c**) Illustration of the model neuron and compartments assayed for SYT7α-HaloTag colocalization. (**d**) Bar graph showing changes in colocalization of SYT7α-HaloTag/JF549 and labeled organelles (M6PR and sortilin label post-Golgi vesicles). Quantified by taking the difference of the PCC between DAPT-treated and control neurons in each condition. Values are means +/- error propagated SEM from at least three separate experiments for each condition. (**e**) Representative in-gel fluorescence of the protein extracted from rat cortical neurons transduced with SYT7α-HaloTag and pulse-chased with JF635 at 13 DIV under control conditions and when grown in 0.5 μM DAPT. Cultures were labeled with JF635 at 13 DIV and then robustly washed with conditioned media. The disappearance of labeled SYT7α-HaloTag/JF635 from the gel can be used to calculate protein half-life. Control SYT7α-HaloTag/JF635 runs between 75 and 100 kDa, while DAPT-treated SYT7α-HaloTag/JF635 runs slightly higher because cleavage of the amino-terminus is blocked. Trichloroethanol (TCE) staining was used as a loading control. (**f**) Normalized intensity of SYT7α-HaloTag/JF635 plotted as the fraction of total control SYT7α-HaloTag/JF635 against days post-wash. Values are means +/- SEM from three independent experiments. Single exponential functions were fitted to control (black) and DAPT (red) conditions. The tau for control SYT7α-HaloTag/JF635 is 9.5 days, while the tau for DAPT-treated SYT7α-HaloTag/JF635 is 3.3 days.

The online version of this article includes the following figure supplement(s) for figure 6:

**Figure supplement 1.** Treatment with DAPT does not recapitulate SYT7KO synaptic phenotypes in vitro.

immunoblot analysis (*Figure 7—figure supplement 1g*) (tittered to a similar expression as the SYT7 WT construct that achieved rescue). First, we found that expression of untagged SYT7α rescues synaptic depression (*Figure 7d*). We further observed that the PM- and endo-lysosomal-targeted constructs both also rescue synaptic depression, while SV-targeted SYT7α does not (*Figure 7e*). This is rather remarkable because SV-targeted SYT7α is present at the site of exocytosis. The observation that this construct does not rescue the KO phenotype emphasizes the importance of precise SYT7α localization. Confidence intervals for the difference between total active synapses throughout the stimulus train are shown in *Figure 7f*, which provides a compact means to visualize all pair-wise comparisons. By quantifying the first two stimuli from the HFS experiments, we calculated the 20 Hz PPF ratio. Using the same methods as in *Figure 1e–f*, we plotted the two components of facilitation, with active synaptic sites on the y-axis and peak iGluSnFR changes in fluorescence on the x-axis. Here, we see that the WT PPF ratio is positive and clusters with full-length SYT7α, PM-, and LAMP1-SYT7α rescue constructs; in contrast, SV-targeted SYT7α failed to rescue PPF (*Figure 7g*). Examining asynchronous release over a train, both SYT7α and PM-SYT7α rescue asynchronous release in SYT7KO neurons, but the LAMP1-SYT7α construct does not, even though it rescues other release modes. Strikingly, when targeted to SVs, SYT7α unexpectedly promoted synchronous release instead of asynchronous release (*Figure 7h*). We also plotted all conditions tested as the synchronous fraction against stimulation number (*Figure 7—figure supplement 1h*), where best fit lines are shown for clarity. This plot illustrates that as the number of successive stimuli increase, the fraction of asynchronous release increases as well.

To summarize the three phenotypes described here, with respect to the rescue constructs, we plotted the average asynchronous fraction of release on the y-axis and depression on the x-axis, and the PPF ratio was encoded in the size of each point in the graph (largest size is the highest PPF ratio, smaller size represents lower or negative PPF ratio) (*Figure 7i*). Only the PM-targeted SYT7α construct rescued all investigated phenotypes. Interestingly, the SV-retargeted SYT7α not only failed to rescue asynchronous release but actually promoted synchronous release. While the underlying mechanism is unclear, this construct may provide a novel tool to tune synchronous release at central synapses.

## Discussion

SYT7 is broadly expressed in the brain but a consensus regarding its precise function in neurons remains the subject of considerable debate. The initial report, in which synaptic function was examined electrophysiologically using constitutive KO mouse lines, concluded that SYT7 played no role in SV exocytosis or synaptic function (*Maximov et al., 2008*). Later, upon the application of more than one stimulus, various deficiencies in SYT7KO neurons were reported; these included reductions in asynchronous release (*Wen et al., 2010*; *Bacaj et al., 2013*), enhanced synaptic depression (*Liu et al., 2014a*), or a loss of PPF (*Jackman et al., 2016*). While deficiencies in all three of these functions were observed at a specific cerebellar synapse in KO mice, only subsets of these functions

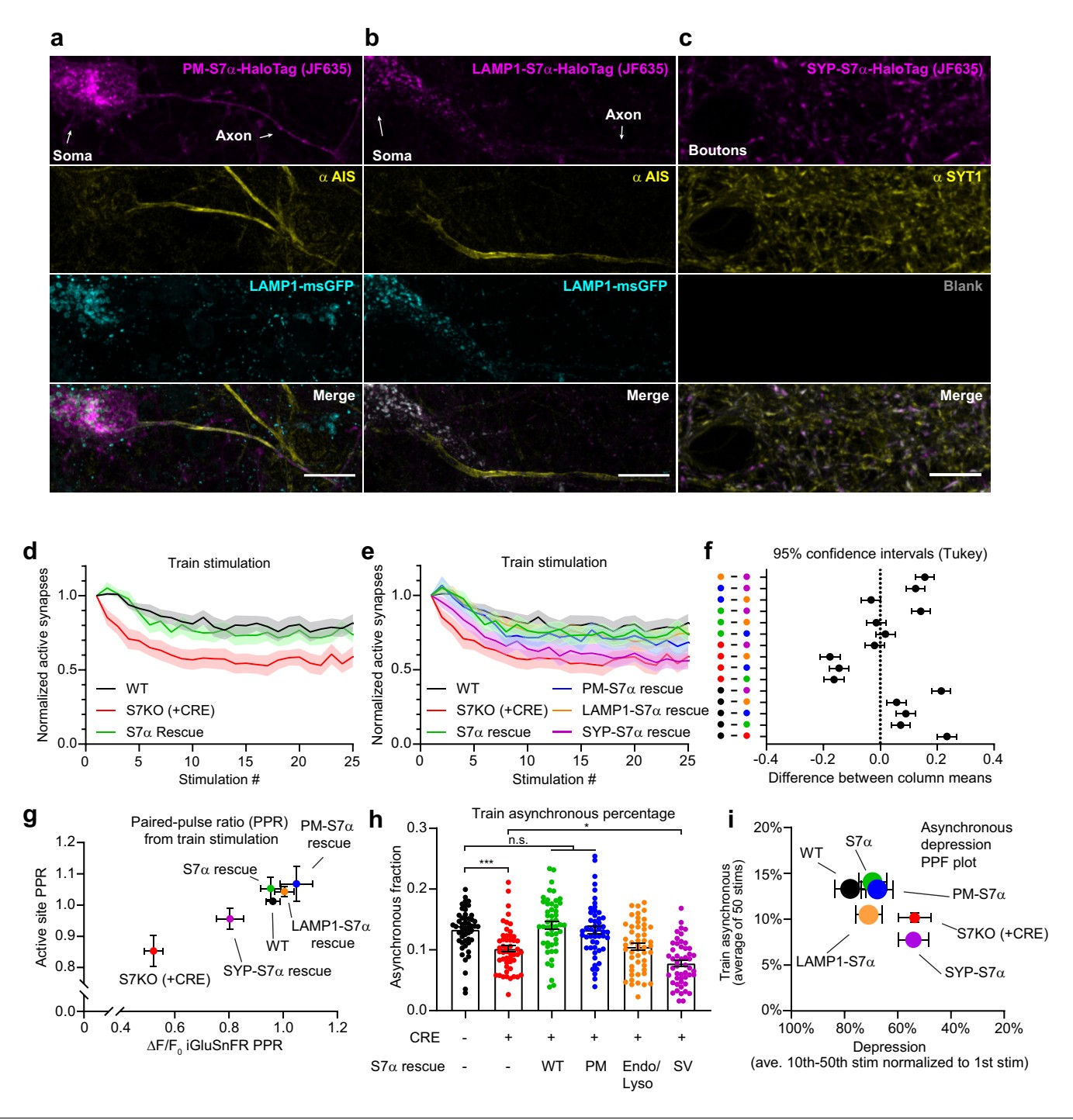

**Figure 7.** Dissociating discrete SYT7 functions via protein retargeting. (**a–c**) Representative super-resolution maximum z-projection of rat hippocampal neurons transduced with (**a**) a plasma membrane-targeted synaptotagmin (SYT)7α, [PM-SYT7α-HaloTag (magenta)], plus LAMP1-msGFP (cyan), (**b**) a lysosome-targeted SYT7α, [LAMP1-SYT7α-HaloTag (magenta)], plus LAMP1-msGFP (cyan), and (**c**) a synaptic vesicle-targeted SYT7α, [SYP-SYT7α-HaloTag (magenta)]. Neurons were fixed and stained with HTL-JF635 (**a–c**), anti-pan-neurofascin (yellow, **a, b**), and anti-SYT1 (yellow, **c**) antibodies. For panel (**c**), a blank image is included to preserve the layout. For panels (**a**) and (**b**), SYT7α constructs were sparsely transduced to better examine localization. Scale bars = 10 μm. (**d**) Depression plot, showing the fraction of active synapses (synapses releasing peak glutamate above baseline, >4 SD above noise) as a function of stimulation number during high-frequency stimulation (HFS). Values are means (solid line) +/- SEM (shaded error), WT (black, n = 15), SYT7KO (red, n = 13), and SYT7α rescue (green, n = 15) from three independent experiments; SYT7KO vs SYT7α rescue is ****p<0.0001 by two-way analysis of variance (ANOVA) comparing genotypes. (**e**) Depression plot from panel (**d**) but with SYT7α rescue constructs included. Values are means (solid line) +/- SEM (shaded error), PM-SYT7α rescue (blue, n = 15), LAMP1-SYT7α rescue (orange, n = 15), and SYP-SYT7α rescue (purple, n

*Figure 7 continued on next page*

*Figure 7 continued*

= 15) from three independent experiments. (**f**) Multiple comparison confidence interval (95% CI) plot from data in panel (**e**). Plot was generated from two-way ANOVA comparing the predicted mean difference between genotypes of normalized active synapses. Comparisons with errors including zero are not statistically different. Total summary statistics are included in *Figure 7—source data 1*. (**g**) An X-Y plot of paired-pulse ratio (PPR) generated at 20 Hz (from first two pulses of HFS). Values are means +/- SEM, where X values are the ratio of the change in glutamate release ($\Delta F/F_0$ iGluSnFR peaks) and Y values are the fraction of regions of interest (ROIs) releasing glutamate (active sites) from wild-type (WT) (black), SYT7KO (red), and SYT7α rescue (green), PM-SYT7α rescue (blue), LAMP1-SYT7α rescue (orange), and SYP-SYT7α rescue (purple). (**h**) Train asynchronous release (peak release recorded between 10 ms and 50 ms post-stimulus) of WT and SYT7KO vs the labeled rescue constructs. Values are means +/- SEM and are the average asynchronous values from each stimulus during a 50 action potential (AP) (20 Hz) HFS; so n = 50 for each group. All comparisons and summary statistics are provided in *Figure 7—source data 2*, and only some are labeled on the graph for presentations sake; p-values are as follows: ***p = 0.001, *p = 0.0147, by one-way ANOVA with Holm-Sidak's multiple comparisons test. (**i**) Summary X-Y plot illustrating different magnitudes of rescue for three of the proposed functions of SYT7. Values are means +/- SEM, where X values represent depression percentage (release from $10^{th}$ to $50^{th}$ stimulation normalized to first) and Y values are the average asynchronous percentage of each genotype during the HFS train. The size of each dot reflects the relative magnitude of each PPR, normalized on a scale from the largest, 10 au (most paired-pulse facilitation (PPF)), to the smallest, 1 au (least PPF). The online version of this article includes the following source data and figure supplement(s) for figure 7:

**Source data 1.** Total summary statistics from multiple comparison confidence interval (95% CI) plot from data in panel (e).
**Source data 2.** Statistic summary using one-way ANOVA with Holm-Sidak's multiple comparisons test for quantification of train asynchronous release.
**Figure supplement 1.** Extended analysis of SYT7 chimeric rescue constructs.

appeared to be disrupted at other kinds of synapses (*Chen et al., 2017*; *Turecek and Regehr, 2018*). In an effort to unify the current thinking concerning SYT7 function, we carefully examined dissociated mouse hippocampal neurons using an optical reporter for glutamate, iGluSnFR. We focused on this preparation because it is a ubiquitous model system in the field, and it allows for tractable investigation of the underlying molecular and cellular mechanisms.

Using iGluSnFR, we detected a small but significant change in asynchronous release from single stimuli between WT and SYT7KO synapses (*Figure 1*); otherwise, there was no apparent change in the amplitude of release evoked by a single stimulus. During paired pulse measurements, we observed that SYT7KO neurons had reduced glutamate transients following the second stimulus, and thus failed to facilitate. Optical detection of release allowed us to further explore the nature of facilitation. We found that in WT synapses, PPF is due to enhanced glutamate release from already active synapses and not from an additional activation of previously silent synapses (i.e., recruitment). During HFS trains, again after the initial stimulus, glutamate release was reduced (resulting in faster and deeper depression) in SYT7KO neurons, and the number of active synapses also decreased to a greater extent, as compared to WT neurons, throughout the train (*Figure 2*). Measuring release during HFS via conventional whole-cell patch clamp produces a train of responses that fail to decay to baseline. This charge transfer component is termed tonic transmission and is thought to arise from either (a) an accumulation of glutamate during HFS, (b) extra-synaptic glutamate 'spill-over', or (c) asynchronously released glutamate. Using iGluSnFR to monitor release during HFS, we recorded glutamate release from individual synapses, and we conclude that 'tonic' transmission results from an increasing fraction of asynchronously released SVs as the stimulus train progresses. We suggest that this is an activity-dependent form of more slowly released SVs, and that this mode of asynchronous release is decreased at SYT7KO synapses (*Figure 2*). Importantly, our reasoning stems from comparisons between averaged iGluSnFR traces with individual iGluSnFR ROIs; in individual traces, during steady-state release, iGuSnFR signals from individual ROIs decay to baseline, whereas averaged iGluSnFR signals do not, strongly supporting asynchronous release as a driver of increased baseline fluorescence in the averaged traces. However, because we are employing the low-affinity iGluSnFR, there may be 'residual' glutamate that electrophysiological measurements detect, but iGluSnFR does not. Additionally, quantal analysis during HFS trains revealed a switch from multiquantal release early in the train to uniquantal release late in the train. Applying quantal analysis to asynchronous release from WT and SYT7KO neurons, we found that a fraction of asynchronous release was multiquantal in WT neurons and that this was decreased in SYT7KO neurons. Therefore, in the absence of SYT7, not only is the frequency of release (after the initial stimulus) for a given neuron fewer in number, but when release does happen, it is decreased in magnitude owing to a lower propensity for multiquantal exocytosis.

Having demonstrated that release is universally reduced in SYT7KO neurons after an initial stimulus (asynchronous, PPF, depression), we sought to investigate how this is manifested in the SV cycle using 'zap-and-freeze' EM. This approach revealed that docked vesicles are more severely depleted by both single stimuli and HFS in SYT7KO (*Figure 3*). Hence, SYT7 serves to promote vesicle docking, or to prevent undocking after a single stimulus and during a stimulus train. Interestingly, we also observed decreases in the total number of SVs during HFS that, in WT neurons, recovered within 5 s, but in the SYT7KO, failed to completely recover over the same time frame. This decrease in SV number after HFS may partly explain the SV replenishment defect observed here (*Figure 2*) and elsewhere (*Liu et al., 2014a*; *Chen et al., 2017*). Defining the mechanism of replenishment is a major goal in the synaptic physiology field, and SYT7 clearly plays a key role in this process. Our data revealed that SYT7 may contribute to replenishment by facilitating activity-dependent SV docking or by preventing AP-triggered undocking (*Kusick et al., 2020*). SYT7 may also play a role in SV reformation during an HFS, again because we observed delayed SV reformation after HFS in the SYT7KO synapses. How the docked vesicle and total vesicle pools are decreased, while vesicles nearest the active zone (but not docked) are unaffected, is an interesting question that will be explored in future studies.

By combining optical SV exocytosis methods with 'zap-and-freeze', we can pin-point where the defects in the SV cycle arise in SYT7KO synapses. This leads to a paradox however, because the docking defect and the failure to recover SV number presumably occur in two different areas of the presynaptic zone. We, therefore, examined the trafficking and localization of SYT7. We used a variety of imaging techniques and expression vectors to localize SYT7 in mature neurons and all our experiments supported the conclusion that (1) SYT7 is asymmetrically polarized to the axonal versus the dendritic plasma membrane and (2) a population of SYT7 resides on the lysosomal membrane of inactive and active lysosomes (*Figure 4*). Interestingly, we observed that processing of SYT7 by the γ-secretase complex is required for targeting to axonal and lysosomal membranes. Membrane protein processing by the γ-secretase complex, while originally postulated to be an intramembrane proteasome, is now accepted as a mechanism to regulate the location of membrane proteins (*Kopan and Ilagan, 2004*). Cleavage of SYT7 by γ-secretase, along with this protein's high sensitivity to palmitoylation inhibitors, makes it an atypical peripheral membrane protein and may afford SYT7 with the ability to quickly transfer between compartments in the presynapse during sustained stimulation, irrespective of the sorting of SV membrane proteins. Support for this idea stems from (1) the speed with which SYT7α is depalmitoylated in the presence of 2-BP, which suggests active and robust palmitoylation/depalmitoylation cycling, its (2) stabilization and (3) axonal enrichment by γ-secretase processing (*Figure 6*). Support against this idea comes from our WT γ-secretase inhibitor experiments. If γ-secretase cleavage is absolutely needed for SYT7 function, then applying inhibitors to WT neurons should phenocopy the SYT7KO, but this was not observed. However, we also showed that SYT7α transits through the PM and that during inhibitor (DAPT) treatment, a detectable amount of SYT7α was still present on axons. So, it is plausible that there is enough SYT7 at the PM to sustain synaptic function in the γ-secretase inhibitor experiments. Interestingly, synapses lacking components of the γ-secretase complex, namely presenilin, have strikingly similar phenotypes to SYT7KO synapses, specifically enhanced depression, and reduced PPF (*Zhang et al., 2009*). Indeed, *Barthet et al., 2018* recently described a role for the γ-secretase complex in the regulation of SYT7 through an interaction with APP. Our observations are not mutually exclusive as we found γ-secretase to positively regulate SYT7, while *Barthet et al., 2018* have proposed a model for how mature SYT7 is negatively regulated via APP. These studies reveal the complex regulation of SYT7 by an unknown acyltransferase and the γ-secretase complex. Future studies will focus on identifying the enzymes responsible for the rapid palmitoylation and depalmitoylation of SYT7, and the impact of disease-associated presenilin mutations on the processing and trafficking of this protein. In sum, these data support the idea that SYT7 exists in axons as a peripheral membrane protein that is anchored to the membrane by labile palmitoylation, which may imbue SYT7 with novel membrane trafficking properties during synaptic activity.

The correct targeting of SYT7α is of critical importance, as shown by our re-targeting experiments. Surprisingly, while the PM-restricted SYT7α construct rescued all functions of WT SYT7α, the SV-targeted construct did not rescue any functions; moreover, this construct served to decrease asynchronous release to below SYT7KO levels. This was unexpected because SV-SYT7α is in the same general area as WT SYT7α, at release sites. Additionally, experiments on another closely

related protein, SYT1, indicate that retargeting this protein from the SV membrane to the PM does not interfere with its function in triggering SV release (*Yao et al., 2011b*). While a portion of SYT7α localized to the endo-lysosomal compartment, and the retargeted LAMP1-SYT7α construct rescued some of the functions of SYT7, we could not define a clear role for lysosomal SYT7 in the SV cycle. Moreover, no alterations in neutral lipids or cholesterol-sensitive SV protein interactions were observed in SYT7KO neurons. We examined this issue because SYT7 was implicated in cholesterol trafficking via lysosome-peroxisome contacts in fibroblasts (*Chu et al., 2015*). Indeed, additional genetic screens exploring cholesterol flux through the lysosome have failed to find a role for SYT7 in this process (*Trinh et al., 2020*). Interestingly, PM- and endo-lysosomal-targeted SYT7α were both able to rescue the synaptic depression and PPF phenotypes that we observed; however, the endo-lysosomal-targeted SYT7α did not rescue asynchronous release (*Figure 7*). This is an important observation because PPF and asynchronous release could have shared a common mechanism; if so, they would not be separable. By retargeting SYT7α, we found that these functions could be disassociated from one another, so these processes are likely to be mechanistically distinct. An additional interpretation of this rescue experiment is that small amounts of the lysosomal construct, which escaped detection, trafficked to the PM, and rescued some phenotypes. Indeed, when transfected with high amounts of DNA (i.e., when greatly overexpressed), the LAMP1-SYT7α construct was detected on the PM. We note that neuronal activity can induce $Ca^{2+}$ release from lysosomes (*McGuinness et al., 2007*) and trigger lysosome exocytosis (*Padamsey et al., 2017*), potentially providing a pathway for the delivery of this chimeric construct to the PM. Regardless of the interpretation, these retargeting experiments reveal a separation between the mechanisms that mediate PPF and depression versus asynchronous release. Again, even more striking was the observation that when re-targeted to SVs, SYT7α did not rescue any SYT7KO synaptic phenotype and instead suppressed asynchronous release, which is the opposite of its normal function. In contrast, when Double C2-like domain-containing protein (DOC2), another protein that plays a role in asynchronous release, was tethered to SVs, it enhanced this slow phase of transmission (*Yao et al., 2011a*). Future studies will address why SYT7 does not behave in the same manner, but the retargeted protein already provides a useful tool to modulate the extent of asynchronous release to potentially alter, for example, reverberatory activity (*Lau and Bi, 2005*). Moreover, because SYT7 functions to promote docking of SVs during activity, to potentially regulate PPF, depression, and asynchronous release, there are likely to be numerous behavioral and coordination abnormalities that have yet to be described. Indeed, a recent report described bipolar disorder symptoms in SYT7KO mice, as well as decreased SYT7 mRNA in the plasma samples of human patients with bipolar disorders (*Shen et al., 2020*).

In summary, we have investigated the trafficking and post-translational processing of SYT7 and identified it as a novel substrate of the γ-secretase complex. SYT7 processing by the γ-secretase complex is required for axonal plasma membrane enrichment and SYT7 protein stability (*Figure 8a*). From the axonal plasma membrane, SYT7 participates in asynchronous release, short-term synaptic plasticity, and SV replenishment. Notably, we have provided, for the first time, SV morphological correlates that help to explain the observed deficits in SYT7KO synapses: SVs in SYT7KO synapses undock during activity to a greater extent than WT controls and fail to efficiently regenerate SVs following HFS (*Figure 8b*). Future work will involve careful 'zap-and-freeze' experiments to follow endosomal intermediates produced during HFS. New advances in correlative light and electron microscopy (CLEM) or fluorescence electron microscopy (fEM) might make it possible to identify the endosomal intermediates that are influenced by SYT7. Moreover, the role of SYT7 and DOC2 (*Yao et al., 2011a*) isoforms in promoting asynchronous release during train stimulation needs further clarification. Clearly, both proteins influence asynchronous release, but their relative contributions remain unresolved. In the experiments reported here, SYT7 promoted late-train asynchronous release, so it is tempting to speculate that both proteins promote asynchronous release but during different phases of the stimulation epoch; perhaps early asynchronous release is DOC2 dependent while late asynchronous release is SYT7 dependent. One model that we will explore is whether SYT7 functions mainly as a promoter of docking and priming while DOC2 mediates the actual triggering of asynchronously released vesicles.

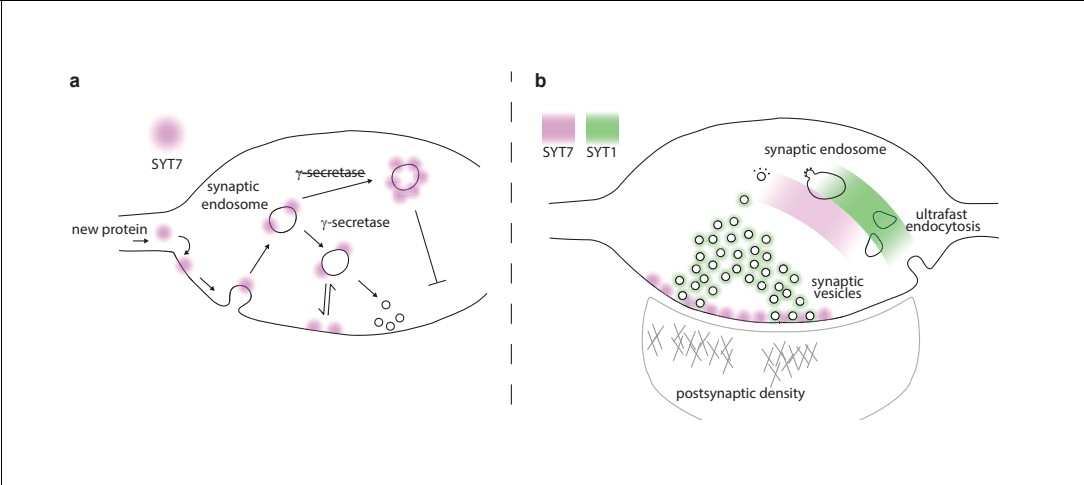

**Figure 8.** Proposed model of presynaptic SYT7. (**a**) Illustration of a nerve terminal in which the location of synaptotagmin (SYT)7 (magenta) at the axonal plasma membrane is dependent on γ-secretase processing and palmitoylation near the transmembrane domain. If palmitoylation is blocked (via drugs or mutations), SYT7 is rapidly degraded. If γ-secretase processing is inhibited, SYT7 mislocalizes to endo-lysosomal intermediate structures. (**b**) A model synapse with the roles and locations of SYT1 and SYT7 indicated by green and magenta shading, respectively. SYT1 is localized to the synaptic vesicle (SV) membrane where it supports docking, priming, drives the formation of the readily releasable pool (RRP), clamps spontaneous release, triggers fast synchronous release, and accelerates endocytosis after exocytosis. The work in the current study revealed that SYT7 physically and functionally localizes to the plasma membrane of the axon, where it plays roles in supporting release during short-term synaptic activity and in reforming SVs. We propose that SYT7 functions, in part, by 'reaching out' to bind SVs to regulate docking in an activity-dependent manner, to control aspects of short-term plasticity.

# Materials and methods

## Key resources table

| Reagent type (species) or resource | Designation | Source or reference | Identifiers | Additional information |
|---|---|---|---|---|
| Biological sample (*Rattus norvegicus*) | Primary rat hippocampal neurons | Envigo | Sprague Dawley | |
| Biological sample (*Mus musculus*) | Primary mouse hippocampal neurons | Jackson Labs | syt7^tm1Nan | *Chakrabarti et al., 2003* |
| Biological sample (*Mus musculus*) | Primary mouse hippocampal neurons | *Codner et al., 2018* | Syt7-TM1C-EM4-B6N | |
| Cell line (*Homo sapiens*) | HEK293T, Kidney epithelial | ATCC | CRL-11268 | |
| Recombinant DNA reagent | FUGW (plasmid) | Addgene | Addgene plasmid # 14883; http://n2t.net/addgene: 14883; RRID:Addgene_14883 | Lentivirus backbone |
| Recombinant DNA reagent | pEF-GFP (plasmid) | Addgene | Addgene plasmid # 11154; http://n2t.net/addgene:11154; RRID:Addgene_11154 | pEF backbone |
| Recombinant DNA reagent | pAAV.hSynapsin.SF-iGluSnFR.S72A (plasmid) | Addgene | Addgene plasmid # 106176; https://www.addgene.org/106176/; RRID:Addgene_106176 | Low-affinity iGluSnFR |
| Recombinant DNA reagent | pHTC HaloTag (plasmid) | Promega (G7711) | pHTC HaloTag CMV-neo Vector (Promega; G7711) | HaloTag |

*Continued on next page*

*Continued*

| Reagent type (species) or resource | Designation | Source or reference | Identifiers | Additional information |
|---|---|---|---|---|
| Recombinant DNA reagent | pLenti-hSynapsin-CRE-WPRE (plasmid) | Addgene | Addgene plasmid # 86641; http://n2t.net/addgene:86641; RRID:Addgene_86641 | CRE vector |
| Recombinant DNA reagent | pORANGE (plasmid) | Addgene | Addgene plasmid # 131471; http://n2t.net/addgene:131471; RRID:Addgene_131471 | pORANGE backbone |
| Recombinant DNA reagent | pF(UG) hSyn SYT7α (plasmid) | This study | Addgene | 'Materials and methods' section |
| Recombinant DNA reagent | pF(UG) hSyn PP-HaloTag-SYT7α (plasmid) | This study | Addgene | 'Materials and methods' section |
| Recombinant DNA reagent | pF(UG) hSyn SYT7α-HaloTag (plasmid) | This study | Addgene | 'Materials and methods' section |
| Recombinant DNA reagent | pF(UG) hSyn SYT7α-HaloTag TMD Cys-Ala (plasmid) | This study | Addgene | 'Materials and methods' section |
| Recombinant DNA reagent | pF(UG) hSyn SYT7α-HaloTag P2A PM-msGFP (plasmid) | This study | Addgene | 'Materials and methods' section |
| Recombinant DNA reagent | pF(UG) hSyn SYP-ΔTMD SYT7α (plasmid) | This study | Addgene | 'Materials and methods' section |
| Recombinant DNA reagent | pF(UG) hSyn SYP-ΔTMD SYT7α-HaloTag (plasmid) | This study | Addgene | 'Materials and methods' section |
| Recombinant DNA reagent | pF(UG) hSyn PM-SYT7αΔTMD (plasmid) | This study | Addgene | 'Materials and methods' section |
| Recombinant DNA reagent | pF(UG) hSyn PM-SYT7αΔTMD-HaloTag (plasmid) | This study | Addgene | 'Materials and methods' section |
| Recombinant DNA reagent | pF(UG) hSyn Lamp1-SYT7αΔTMD (plasmid) | This study | Addgene | 'Materials and methods' section |
| Recombinant DNA reagent | pF(UG) hSyn Lamp1-SYT7αΔTMD-HaloTag (plasmid) | This study | Addgene | 'Materials and methods' section |
| Recombinant DNA reagent | pF(UG) U6-SYT7 sgRNA 777 Halo Tag (plasmid) | This study | Addgene | 'Materials and methods' section |
| Recombinant DNA reagent | pF(UG) hSyn spCas9 (plasmid) | This study | Addgene | 'Materials and methods' section |
| Recombinant DNA reagent | pF(UG) CamKII sf iGluSnFR S72A (plasmid) | This study | Addgene | 'Materials and methods' section |
| Recombinant DNA reagent | pF(UG) hSyn SYP-mRuby3 (plasmid) | This study | Addgene | 'Materials and methods' section |
| Recombinant DNA reagent | pF(UG) hSyn LAMP1-msGFP (JV012) (plasmid) | This study | Addgene | 'Materials and methods' section |
| Recombinant DNA reagent | pEF-GFP (plasmid) | This study | Addgene | 'Materials and methods' section |
| Recombinant DNA reagent | pEF-mRuby3 (plasmid) | This study | Addgene | 'Materials and methods' section |
| Software, algorithm | SynapsEM | | https://github.com/shigekiwatanabe/SynapsEM. | |

*Continued on next page*

*Continued*

| Reagent type (species) or resource | Designation | Source or reference | Identifiers | Additional information |
|---|---|---|---|---|
| Sequence-based reagent | SYT7 sgRNA 777 | This paper | sgRNA | CACCAGCTGA AAGCCTGAGA |
| Antibody | Anti-SYT1 (mouse monoclonal) | DSHB | mAB 48; RRID:AB_2199314 | IB (1:1000) ICC (1:100) |
| Antibody | Anti-SYP (guinea pig polyclonal) | SySy | 101 004; RRID:AB_1210382 | IB (1:1000) ICC (1:500) |
| Antibody | Anti-SYT7 (rabbit polyclonal) | SySy | 105 173; RRID:AB_887838 | IB (1:1000) ICC (1:100) |
| Antibody | Anti-HaloTag (mouse monoclonal) | Promega | G9211; RRID:AB_2688011 | IB (1:1000) |
| Antibody | Anti-pan-neurofascin (mouse monoclonal) | NIH NeuroMab | 75–172; RRID:AB_2282826 | ICC (1:200) |
| Antibody | Anti-SEC61A (rabbit monoclonal) | Abcam | ab183046; RRID:AB_2620158 | ICC (1:100) |
| Antibody | Anti-GM130 (mouse monoclonal) | BD Biosciences | 610822; RRID:AB_398141 | ICC (1:100) |
| Antibody | Anti-TGN38/46 (mouse monoclonal) | Abcam | ab2809; RRID:AB_2203290 | ICC (1:20) |
| Antibody | Anti-EEA1 (rabbit polyclonal) | Abcam | ab2900; RRID:AB_2262056 | ICC (1:50) |
| Antibody | Anti-M6PR (mouse monoclonal) | Abcam | ab2733; RRID:AB_2122792 | ICC (1:100) |
| Antibody | Anti-sortilin (rabbit polyclonal) | Abcam | ab16640; RRID:AB_2192606 | ICC (1:100) |
| Antibody | Anti-HaloTag (rabbit polyclonal) | Promega | G9281; RRID:AB_713650 | ICC (1:500) |
| Chemical compound, drug | HTL-JF549 | Janelia Research Campus/ HHMI | | Luke Lavis Lab |
| Chemical compound, drug | HTL-JF635 | Janelia Research Campus/ HHMI | | Luke Lavis Lab |
| Chemical compound, drug | HTL-JF646 | Janelia Research Campus/ HHMI | | Luke Lavis Lab |
| Chemical compound, drug | HTL-JF549i | Janelia Research Campus/ HHMI | | Luke Lavis Lab |
| Chemical compound, drug | Prosense 680 | PerkinElmer | NEV10003 | 0.5 mM |

## Cell culture

Sprague Dawley rat hippocampal and cortical neurons were isolated at E18 (Envigo). Mouse hippocampal neurons from the *Syt7* floxed mouse strains Syt7-TM1C-EM4-B6N (*Codner et al., 2018*) and *Syt7^tm1Nan^* (*Chakrabarti et al., 2003*) were isolated at P0 and prepared using a procedure previously described in *Vevea and Chapman, 2020*. Briefly, hippocampal neurons were dissected, trypsinized (Corning; 25–053 CI), triturated, and plated on glass coverslips (Warner instruments; 64–0734 (CS-18R17)) coated with poly-D-lysine (Thermofisher; ICN10269491) and Engelbreth-Holm-Swarm (EHS) laminin (Thermofisher; 23017015). Neurons were grown in Neurobasal-A (Thermofisher; 10888–022) medium supplemented with B-27 (2%, Thermofisher; 17504001), Glutamax (2 mM, Gibco; 35050061), and pen/strep before experiments. For high-pressure freezing and EM, cell cultures were prepared on 6 mm sapphire disks (Technotrade), mostly as previously described (*Kusick et al.,*

*2020*). For two of the three experiments/cultures, genotyping was performed after hippocampal dissection, using cortices, with hippocampi left in neurobasal media-A (NBM-A) before switching to papain after, while in the third, genotyping was performed using tail clips. Before use, sapphire disks were carbon-coated with a '4' to indicate the side that cells were cultured on. Health of the cells, as indicated by de-adhered processes, floating dead cells, and excessive clumping of cell bodies, was assessed regularly, as well as immediately before experiments. All EM-related experiments were performed between 13 and 17 DIV. All other experiments were performed between 14 and 23 DIV. For virus preparation, HEK293T cells (ATCC) were cultured following ATCC guidelines and were tested for mycoplasma contamination using the Universal Mycoplasma Detection Kit (ATCC; 30–1012K); HEK293T cells were validated using short tandem repeat profiling by ATCC (ATCC; 135-XV) within the previous year.

## Lentivirus production and use

Lentivirus production was performed as described previously (*Vevea and Chapman, 2020*). Lentiviral constructs were all subcloned into the FUGW transfer plasmid (FUGW was a gift from David Baltimore (Addgene plasmid # 14883; http://n2t.net/addgene:14883; RRID:Addgene_14883)) (*Lois et al., 2002*). We previously replaced the ubiquitin promoter with the CAMKII promoter or human synapsin I promoter (*Kügler et al., 2003*; *Vevea and Chapman, 2020*). Lentiviruses that express CRE or iGluSnFR were added to neuronal cultures at 1 DIV. SYT7α rescue and other constructs that were used to mark organelles were added at 5–6 DIV.

## Plasmid construction

Two types of plasmids were used in this study. One was our previously modified lentivirus backbone of choice derived from FUGW (*Vevea and Chapman, 2020*), and the other was based on pEF-GFP, excising the green fluorescent protein (GFP) and substituting our own various inserts (pEF-GFP was a gift from Connie Cepko (Addgene plasmid # 11154; http://n2t.net/addgene:11154; RRID:Addgene_11154)) (*Matsuda and Cepko, 2004*). The low-affinity glutamate sensor iGluSnFR S72A was polymerase chain reaction (PCR) amplified from pAAV.hSynapsin.SF-iGluSnFR.S72A, which was a gift from Loren Looger (Addgene plasmid # 106176; https://www.addgene.org/106176/; RRID:Addgene_106176) (*Marvin et al., 2018*), and subcloned into our lentivirus transfer plasmid (CamKII promoter) along with the addition of membrane trafficking motifs to promote PM localization as done previously for the original iGluSnFR variant (*Vevea and Chapman, 2020*). The SYT7α rescue constructs were assembled using PCR splicing with overlap extension (SOE) and subcloned into our modified FUGW transfer plasmid. To retarget SYT7α, we fused the cytosolic domain (including juxtamembrane linker) to various protein domains. For PM-targeted SYT7α, we used a preprolactin signal sequence (PP) fused to a CD4 TMD along with an adjacent Golgi export sequence (*Parmar et al., 2014*) amended to the amino-terminus of SYT7α. For endo-lysosomal targeting, we added the cytosolic domain of SYT7α to the carboxy-terminus of the LAMP1 protein, and for synaptic vesicle targeting, the cytosolic domain of SYT7α was fused to the carboxy-terminus of synaptophysin. For HaloTag fusions, the HaloTag cassette was amplified from the pHTC HaloTag CMV-neo Vector (Promega; G7711) and amended to either the amino- or the carboxy-terminus of SYT7α constructs. If added to the amino-terminus, a preprolactin leader sequence was included to ensure protein translocation across the ER membrane. The fluorescent protein msGFP (*Vevea and Chapman, 2020*) or mRuby3 (*Bajar et al., 2016*) was appended to the carboxy-terminus of LAMP1 for use as a lysosomal marker, synaptophysin as a synaptic vesicle marker, the PM motifs mentioned above for PM targeting, or alone for a cytosolic marker. For CRE expression, we used pLenti-hSynapsin-CRE-WPRE (pLenti-hSynapsin-CRE-WPRE, which was a gift from Fan Wang (Addgene plasmid # 86641; http://n2t.net/addgene:86641; RRID:Addgene_86641)) (*Sakurai et al., 2016*). For HITI of SYT7, we used the protocol developed by *Willems et al., 2020*. Briefly, we cloned SYT7 carboxy gRNA and gRNA-flanked HaloTag into pORANGE (pORANGE cloning template vector was a gift from Harold MacGillavry (Addgene plasmid # 131471; http://n2t.net/addgene:131471; RRID:Addgene_131471)). We were unable to observe HITI events with sparse transfection, and so we split the pORANGE vector into the U6-driven SYT7 gRNA and HaloTag vector and the human synapsin (hSyn)-driven spCas9 lentiviral vector. Using these constructs for complete coverage, we were able to document HITI

events, which demonstrated asymmetric SYT7 localization to the axonal compartment. All original plasmids used in this study are deposited in Addgene, filed under this manuscript.

## SYT7 constructs

pF(UG) hSyn SYT7α
pF(UG) hSyn PP-HaloTag-SYT7α
pF(UG) hSyn SYT7α-HaloTag
pF(UG) hSyn SYT7α-HaloTag TMD Cys- Ala
pF(UG) hSyn SYT7α-HaloTag P2A PM-msGFP
pF(UG) hSyn SYP-ΔTMD SYT7α
pF(UG) hSyn SYP-ΔTMD SYT7α-HaloTag
pF(UG) hSyn PM-SYT7αΔTMD
pF(UG) hSyn PM-SYT7αΔTMD-HaloTag
pF(UG) hSyn Lamp1-SYT7αΔTMD
pF(UG) hSyn Lamp1-SYT7αΔTMD-HaloTag
pF(UG) U6-SYT7 sgRNA 777 HaloTag
pF(UG) hSyn spCas9

## Biosensors

pF(UG) CamKII sf iGluSnFR S72A

## Organelle markers

pF(UG) hSyn SYP-mRuby3
pF(UG) hSyn LAMP1-msGFP (JV012)
pEF-GFP
pEF-mRuby3

## Immunoblot protocol

Immunoblots were performed as described previously (*Vevea and Chapman, 2020*). Primary antibodies used were anti-SYT1 (1:1000, 48) (lab stock; mAB 48; RRID:AB_2199314), anti-SYP (1:1000) (SySy; 101 004; RRID:AB_1210382), anti-SYT7 (1:1000) (SySy; 105 173; RRID:AB_887838), and anti-HaloTag (1:1000) (Promega; G9211; RRID:AB_2688011). Secondary antibodies used were goat anti-mouse IgG2b-HRP (Biorad; M32407; RRID:AB_2536647), goat anti-mouse IgG-HRP (Biorad; 1706516; RRID:AB_11125547), goat anti-rabbit IgG-HRP (Biorad; 1706515; RRID:AB_11125142), and goat anti-guinea pig IgG-HRP (Abcam; ab6908; RRID:AB_955425).

## High-pressure freezing and freeze substitution

Cells cultured on sapphire disks were frozen using an EM ICE high-pressure freezer (Leica Microsystems), exactly as previously described (*Kusick et al., 2020*). The freezing apparatus was assembled on a table heated to 37°C in a climate control box, with all solutions pre-warmed (37°C). Sapphire disks with neurons were carefully transferred from culture medium to a small culture dish containing physiological saline solution (140 mM NaCl, 2.4 mM KCl, 10 mM 4-(2-hydroxyethyl)-1-piperazinee-thanesulfonic acid (HEPES), 10 mM glucose; pH adjusted to 7.3 with NaOH, 300 mOsm). 2,3-Dioxo-6-nitro-1,2,3,4-tetrahydrobenzo[f]quinoxaline-7-sulfonamide disodium salt (NBQX) (3 µM; Tocris) and bicuculline (30 µM; Tocris) were added to the physiological saline solution to block recurrent synaptic activity. $CaCl_2$ and $MgCl_2$ concentrations were 1.2 mM and 3.8 mM, respectively. After freezing, samples were transferred under liquid nitrogen to an EM AFS2 freeze-substitution system at −90°C (Leica Microsystems). Using pre-cooled tweezers, samples were quickly transferred to anhydrous acetone at −90°C. After disassembling the freezing apparatus, sapphire disks with cells were quickly moved to cryovials containing freeze-substitution solutions. For the first two experiments, freeze substitution was performed exactly as previously described (*Kusick et al., 2020*): solutions were 1% glutaraldehyde, 1% osmium tetroxide, and 1% water in anhydrous acetone, which had been stored under liquid nitrogen and then moved to the AFS2 immediately before use. The freeze-

substitution program was as follows: −90℃ for 6–10 hr (adjusted so that substitution would finish in the morning), 5℃ h$^{-1}$ to −20℃, 12 hr at −20℃, and 10℃ h$^{-1}$ to 20℃. For the third experiment, we used a different freeze-substitution protocol that yields a more consistent high-contrast morphology: samples were first left in 0.1% tannic acid and 1% glutaraldehyde at −90℃ for ~36 hr, then washed 5x, once every 30 min, with acetone, and transferred to 2% osmium tetroxide, then run on the following program: 11 hr at −90℃, 5℃ h$^{-1}$ to −20℃, −20℃ for 12 hr, 10℃ h$^{-1}$ to −4℃, then removed from the freeze-substitution chamber and warmed at room temperature for ~15 min before washing.

## Embedding, sectioning, and transmission electron microscopy

Embedding and sectioning were performed exactly as previously described (*Kusick et al., 2020*). For ultramicrotomy, 40 nm sections were cut. Sections on single-slot grids coated with 0.7% pioloform were stained with 2.5% uranyl acetate, then imaged at 80 kV on the x93,000 setting on a Phillips CM 120 transmission electron microscope equipped with an AMT XR80 camera run on AMT Capture v6 for the first experiment, and for the other two experiments, samples were imaged on a Hitachi 7600 transmission electron microscope equipped with an AMT XR50 run on AMT Capture v6. Samples were blinded before imaging. To further limit bias, synapses were found by bidirectional raster scanning along the section at x93,000 or x100,000, which makes it difficult to 'pick' certain synapses, as a synapse usually takes up most of this field of view. Synapses were identified by a vesicle-filled presynaptic bouton and a postsynaptic density. Postsynaptic densities are often subtle in our samples, but synaptic clefts were also identifiable by (1) their characteristic width, (2) the apposed membranes following each other closely, and (3) vesicles near the presynaptic active zone. 125–150 micrographs per sample of anything that appeared to be a synapse were taken without close examination.

## Electron microscopy image analysis

EM image analysis was performed as previously described (*Kusick et al., 2020*). All the images from a single experiment were randomized for analysis as a single pool. Only after this randomization were any images excluded from analysis, either because they appeared to not contain a bona fide synapse or the morphology was too poor for reliable annotation. The PM, the active zone, docked SVs, and all SVs in the bouton were annotated in ImageJ using SynapsEM plugins (*Watanabe et al., 2020*) [https://github.com/shigekiwatanabe/SynapsEM copy archived at swh:1:rev:11a6227cd5951bf5e077cb9b3220553b506eadbe (*Watanabe, 2021*)]. The active zone was identified as the region of the presynaptic PM with the features described above for identifying a synapse. Docked vesicles were identified by their membrane appearing to be in contact with the PM at the active zone (0 nm from the PM); that is, there are no lighter pixels between the membranes. Vesicles that were not manually annotated as docked but were 0 nm away from the active zone PM were automatically counted as docked when segmentation was quantitated (see below) for data sets counting the number of docked vesicles. Likewise, vesicles annotated as docked were automatically placed in the 0 nm bin of vesicle distances from the PM. To minimize bias and error, and to maintain consistency, all image segmentation, still in the form of randomized files, was thoroughly checked and edited by a second member of the lab. Features were then quantitated using the SynapsEM (*Watanabe et al., 2020*) family of MATLAB (MathWorks) scripts (https://github.com/shigekiwatanabe/SynapsEM). Example electron micrographs shown were adjusted in brightness and contrast to different degrees (depending on the varying brightness and contrast of the raw images), rotated, and cropped in ImageJ before import into Adobe Illustrator.

## In-gel fluorescence assay

SYT7α half-life calculations were determined using data obtained via in-gel fluorescence assays. Rat cortical neurons were transduced with SYT7α-HaloTag expression vectors at 5 DIV. At 13 DIV, neurons were incubated with 100 nM JF635 for 30 min at 37℃. These neurons were then washed three times with conditioned NBM-A, and the final wash was replaced with conditioned NBM-A from sister cultures. Samples were harvested at 13 (post-label day 0), 15, 17, 19, and 21 DIV (post-label day 2, 4, 6, and 8, respectively) and subjected to standard SDS-PAGE. Gels were analyzed using a BioRad Chemidoc MP imager (BioRad) using far red fluorescence excitation and emission filters. Data were quantified by densitometry using Fiji (*Schindelin et al., 2012*).

## Immunocytochemistry

ICC was performed as previously described (*Vevea and Chapman, 2020*). Primary antibodies used were anti-SYT1 (1:100, 48) (lab stock; mAB 48; RRID:AB_2199314), anti-SYP (1:500) (SySy; 101 004; RRID:AB_1210382), anti-SYT7 (1:100) (SySy; 105 173; RRID:AB_887838), anti-pan-neurofascin (1:200) (UC Davis/NIH NeuroMab; 75–172; RRID:AB_2282826), anti-SEC61A (1:100) (Abcam; ab183046; RRID:AB_2620158), anti-GM130 (1:100) (BD Biosciences; 610822; RRID:AB_398141), anti-TGN38/46 (1:20) (Abcam; ab2809; RRID:AB_2203290), anti-EEA1 (1:50) (Abcam; ab2900; RRID:AB_2262056), anti-M6PR (1:100) (Abcam; ab2733; RRID:AB_2122792), anti-sortilin (1:100) (Abcam; ab16640; RRID:AB_2192606), and anti-HaloTag (1:500) (Promega; G9281; RRID:AB_713650). Secondary antibodies used include goat anti-mouse IgG1 IgG-Alexa Fluor 488 (1:1000) (Thermofisher; A-21121; RRID:AB_2535764), goat anti-guinea pig IgG-Alexa Fluor 488 (1:1000) (Thermofisher; A-11073; RRID:AB_2534117), goat anti-mouse IgG2a-Alexa Fluor 488 (1:1000) (Thermofisher; A-21131; RRID:AB_2535771), goat anti-rabbit IgG-Alexa Fluor 488 (1:1000) (Thermofisher; A-11008; RRID:AB_143165), goat anti-rabbit IgG-Alexa Fluor 546 (1:1000) (Thermofisher; A-11035; RRID:AB_2534093), goat anti-mouse IgG2a-Alexa Fluor 546 (1:1000) (Thermofisher; A-21133; RRID:AB_2535772), goat anti-mouse IgG2a-Alexa Fluor 647 (1:1000) (Thermofisher; A-21241; RRID:AB_2535810), and goat anti-mouse IgG2b-Alexa Fluor 546 (1:1000) (Thermofisher; A-21143; RRID:AB_2535779). Images for *Figures 3*, *4*, *5*, *6* and *7* were acquired on a Zeiss LSM 880 with a x63 1.4 NA oil immersion objective using the Airyscan super-resolution detector and deconvolved using automatic Airyscan settings. For *Figure 6*, identical laser and gain settings were used in each condition. The same linear brightness and contrast adjustments were applied to all conditions. Images in *Figure 4—figure supplement 3e* were acquired using HILO to detect endogenously tagged SYT7 on an Olympus IX83 inverted microscope equipped with a cellTIRF 4Line excitation system and using an Olympus x60/1.49 Apo N objective with an Orca Flash4.0 CMOS camera (Hamamatsu Photonics).

## Janelia Fluor dye usage

HTL-conjugated JF dyes were graciously provided by Luke Lavis from the Janelia Research Campus. We made use of JF549, JF635, JF646, and JF549i. For protein localization in live neurons or HEK293T cells, cultures were incubated with 100 nM JF549, JF635, or JF646, for 30–60 min at 37°C. Cultures were washed once and imaged. For ICC experiments that required a JF label, a JF dye was added to the primary antibody mix and incubated overnight at 4°C. For the experiment in *Figure 5e* and S5e, we used neurons transduced with HaloTag-SYT7α and incubated with 1 nM of the impermeant JF549i dye for 2 days at 37°C. Incubation for 4 days showed no detectable nonspecific uptake of this dye. This dye allowed us to determine if the amino-terminal portion of SYT7α transited through the PM before being cleaved by γ-secretase intracellularly.

## Live-cell imaging (excluding iGluSnFR)

HEK293T (*Figure 7—figure supplement 1d*) and primary rodent neuronal (*Figure 4b,d*, S3a, c, S4c, and S8c) cultures were transiently transfected with various constructs (pEF and pF(UG) hSyn based) using Lipofectamine LTX reagent with PLUS reagent (Thermofisher; A12621) and imaged using a Zeiss LSM880 with Airyscan confocal microscope. Coverslips containing HEK293T or neuronal cultures were placed in standard imaging media (ECF (extracellular fluid)) consisting of 140 mM NaCl, 5 mM KCl, 2 mM $CaCl_2$, 2 mM $MgCl_2$, 5.5 mM glucose, 20 mM HEPES (pH 7.3), B27 (Gibco), and glutamax (Gibco), loaded onto the microscope, and maintained at physiological temperature (~35°C) with humidity controls to prevent evaporation. Cells were imaged in Fast Airyscan mode and processed with automatic Airyscan deconvolution settings after image acquisition. Neurons were incubated with 0.5 mM Prosense 680 (PerkinElmer; NEV10003) overnight (10–12 hr) to reveal active lysosomes. Incubation for longer time periods, up to 5 days, was tested, revealing no obvious cytotoxicity or improvement in the number or fluorescence magnitude of Prosense 680 signal.

## iGluSnFR imaging and quantification

Synaptic vesicle exocytosis and glutamate release were monitored via iGluSnFR imaging as previously described (*Vevea and Chapman, 2020*), with the following modifications; Images were acquired at 2x2 binning using low-affinity iGluSnFR (S72A mutation) (*Marvin et al., 2018*), and imaging media contained 2 mM $Ca^{2+}$. For single stimuli imaging, 150 frames were collected at 10 ms

exposures (1.5 s total), and a single field stimulus was triggered at half a second after the initial frame. For paired-pulse imaging, two field stimuli were triggered 50 (20 Hz), 100 (10 Hz), 200 (5 Hz), or 500 ms (2 Hz) apart. As above, 150 frames were collected using a 10 ms exposure (1.5 s total), with stimuli after 500 ms baseline. For high-frequency train stimulation (HFS), 50 stimuli were triggered from field depolarizations at 20 Hz (2.5 s), and 350 frames with 10 ms exposures were collected (3.5 s), with the HFS start after 500 ms baseline. Glutamate peaks were recorded when the signal was >4x SD of the noise. This was used as the threshold to identify ROIs that released glutamate and for quantification regarding active synapses.

### Colocalization quantification

Pearson correlation coefficient was quantified as described previously (*Vevea and Chapman, 2020*), using Fiji for ImageJ and Just Another Colocalization Plugin (JACoP) (*Bolte and Cordelières, 2006*). Groups were quantified, and the simple difference between DAPT and control conditions, with error propagated, is displayed in *Figure 6d*.

### Compounds and chemicals

Protease inhibitors used were DAPT (Apexbio; GSI-IX), GI254023X (MedChemExpress; HY-199956), TAPI-1 (ApexBio; B4686), and Verubecestat (ApexBio; MK-8931). The palmitoylation inhibitor used was 2-BP (Sigma; 238422–10G).

### Statistics

Exact values from experiments and analysis, including number of data points (n) and number of trials for each experiment, are listed in the figure legends. EM data are from 2554 total images (2-D synaptic profiles) from three experiments (true biological replicates, different cultures/litters frozen on different days and each imaged and analyzed separately as their own batch). Analysis was done with GraphPad Prism 8.4.3 (GraphPad Software Inc). Data sets were tested for normality using the Anderson-Darling test; if normal, parametric statistical methods were used to analyze data, and if not normal, nonparametric methods were used for analysis.

## Acknowledgements

We would like to thank the Chapman lab members for valuable discussions related to this manuscript and C Greer specifically for valuable edits. We would also like to thank DT Larson, JH Rinald, and K Itoh for excellent technical assistance. This study was supported by grants from the NIH (MH061876 and NS097362 to ERC and NS111133-01 and NS105810-01A1 to SW). JDV was supported by a postdoctoral fellowship from the NIH F32 NS098604 and a Warren Alpert Distinguished Scholars Fellowship award. GFK was supported by a grant from the National Institutes of Health to the Biochemistry, Cellular and Molecular Biology program of the Johns Hopkins University School of Medicine (T32 GM007445) and is a National Science Foundation Graduate Research Fellow (2016217537). The EM ICE high-pressure freezer was purchased partly with funds from an equipment grant from the National Institutes of Health (S10RR026445) awarded to SC Kuo. SW was supported by start-up funds from the Johns Hopkins University School of Medicine, Johns Hopkins Discovery funds, Johns Hopkins Catalyst Award, the National Science Foundation (1727260). ERC is an investigator of the Howard Hughes Medical Institute. SW is an Alfred P Sloan fellow, a McKnight Foundation scholar, and a Klingenstein and Simons Foundation scholar.

## Additional information

### Funding

| Funder | Grant reference number | Author |
|---|---|---|
| National Institutes of Health | MH061876 | Jason D Vevea<br>Kevin C Courtney<br>Edwin R Chapman |
| National Institutes of Health | NS097362 | Jason D Vevea<br>Kevin C Courtney |

| | | Edwin R Chapman |
|---|---|---|
| National Institutes of Health | NS098604 | Jason D Vevea |
| National Science Foundation | 1727260 | Grant F Kusick<br>Erin Chen<br>Shigeki Watanabe |
| National Institutes of Health | NS111133-01 | Grant F Kusick<br>Erin Chen<br>Shigeki Watanabe |
| National Institutes of Health | NS105810-01A1 | Grant F Kusick<br>Erin Chen<br>Shigeki Watanabe |
| National Institutes of Health | GM007445 | Grant F Kusick |
| National Science Foundation | 2016217537 | Grant F Kusick |
| National Institutes of Health | S10RR026445 | Shigeki Watanabe |

The funders had no role in study design, data collection and interpretation, or the decision to submit the work for publication.

### Author contributions

Jason D Vevea, Conceptualization, Data curation, Formal analysis, Funding acquisition, Validation, Investigation, Visualization, Methodology, Writing - original draft, Project administration, Writing - review and editing; Grant F Kusick, Data curation, Formal analysis, Validation, Investigation, Visualization, Methodology, Writing - review and editing; Kevin C Courtney, Conceptualization, Data curation, Formal analysis, Methodology, Writing - review and editing; Erin Chen, Formal analysis; Shigeki Watanabe, Resources, Software, Supervision, Funding acquisition, Project administration, Writing - review and editing; Edwin R Chapman, Conceptualization, Resources, Supervision, Funding acquisition, Writing - original draft, Project administration, Writing - review and editing

### Author ORCIDs

Jason D Vevea ⓘ https://orcid.org/0000-0002-3068-973X
Grant F Kusick ⓘ https://orcid.org/0000-0002-4312-3495
Kevin C Courtney ⓘ https://orcid.org/0000-0003-1315-4917
Erin Chen ⓘ http://orcid.org/0000-0002-6426-1758
Shigeki Watanabe ⓘ https://orcid.org/0000-0001-7580-8141
Edwin R Chapman ⓘ https://orcid.org/0000-0001-9787-8140

### Ethics

Animal experimentation: Animal care and use in this study were conducted under guidelines set by the NIH Guide for the Care and Use of Laboratory Animals handbook. The protocols were reviewed and approved by the Animal Care and Use Committee (ACUC) at the University of Wisconsin, Madison (Laboratory Animal Welfare Public Health Service Assurance Number: A3368-01).

### Decision letter and Author response

Decision letter https://doi.org/10.7554/eLife.67261.sa1
Author response https://doi.org/10.7554/eLife.67261.sa2

# Additional files

### Supplementary files

- Source data 1. Full source data and statisitcs.
- Transparent reporting form

## Data availability

Detailed summary statistics are included in the source data 1-7 for figures 1e, 1f, S1b, 3c, 3d, 7f, 7h. Raw blot and gel images are attached as supplementary zip file.

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
