## [Decision Letter]

**Acceptance summary:**

Your work nicely reports the roles of Syt-VII, a ca^2+^ effector associated with various membranes, in multiple steps of synaptic vesicle exocytosis in cultured hippocampal neurons. Using optical GluSnFR imaging and Zap-freeze EM, you present compelling data that Syt-VII regulates activity-dependent synaptic vesicle (SV) docking and replenishment. These data support previous conclusions that were drawn based on the electrophysiological measurements.

**Decision letter after peer review:**

[Editors’ note: the authors submitted for reconsideration following the decision after peer review. What follows is the decision letter after the first round of review.]

Thank you for submitting your work entitled "Synaptotagmin 7 is enriched at the plasma membrane to promote vesicle docking and control synaptic plasticity" for consideration by *eLife*. Your article has been reviewed by 2 peer reviewers, and the evaluation has been overseen by a Reviewing Editor and a Senior Editor. The reviewers have opted to remain anonymous.

We are sorry to say that, after consultation with the reviewers, we have decided that your work will not be considered further for publication by *eLife*.

As you will notice, both reviewers have carefully evaluated your manuscript. They both argue that the first section is confirmatory of what has been previously published. They also agree that the second section is not well developed and argue that a significant amount of work would have to be performed to address the concerns that they raise. Upon discussions they both feel that the manuscript is not suitable for publication in *eLife*. We therefore decided that your manuscript is a better fit for a more specialized journal.

*Reviewer #1:*

Vevea et al. describe an analysis of Synaptotagmin 7 function in neurotransmission in cultured hippocampal neurons. The authors re-capitulate several previously published phenotypes that were found via electrophysiological approaches with optical GluSnFR analysis in the current study. They also employ zap and freeze EM to identify a small defect in SV docking and SV replenishment after high frequency stimulation. The authors then turn to examine SYT7 localization and suggest a role for γ-secretase cleavage of the protein in facilitating its axonal localization. They argue for SYT7 co-localizing with both lysosomes and the plasma membrane, but most of these experimental approaches are quite confusing and are hard to interpret without more careful localization analysis with a broader panel of compartmental markers. Finally, the author use a mistargeting strategy to suggest SYT7 can reside on either lysosomes or the plasma membrane to facilitate the majority of its functional roles in SV trafficking. Again, I find these particular aspects of the study difficult to interpret, as the localization methods do not appear to be very clean and they are strongly confounded by protein levels – one has no idea of the relative amount of SYT7 compared to the normal amounts in each locale, and reduced or enhanced levels could severely confound the strong interpretations the authors have made. Overall, although the 1st half of the work validates some prior observations in the field and extends the work with EM analysis, the second half is quite challenging to interpret and place in an appropriate context.

1. The first several figures of the paper generally recapitulate work from the field, including the authors' own electrophysiological analysis (Liu et al., 2014 *eLife*), on SYT7 knockout mouse lines. The authors find reduced paired-pulse facilitation and a reduction in synaptic vesicle replenishment rate during high frequency stimulation. They also note a very small change in asynchronous release (~3%) in the absence of SYT7. The main advance here is the use of the GluR sniffer approach versus electrophysiology. The drawback, however, is the variability in the optical method, as it depends on the number of release sites imaged per ROI in each culture. The authors did not quantify single synaptic vesicle signals (minis), so the quantal response from GluR sniffer δ f signals at each site seems unknown. Whether 1 or 50 synaptic vesicles are released during each stimulation is unclear without a baseline quantal signal. Therefore, it seems difficult to make any quantitative comments about single stimulation responses across genotypes or even across ROIs within the same genotype. Only changes from the initial response of an ROI in each culture seem robustly quantifiable in terms of an absolute value. This variability is obvious in Figure 2a, where responses to individual stimuli both within the same field of view or different field of views (A-E) bounce around significantly. As such, the value of the imaging approach in cultured neurons over more well-characterized electrophysiological approaches in defined neuronal populations from slices is unclear.

2. The authors previously reported the RRP was not altered using electrophysiology measurements, but it appears their optical and EM approach suggest a different conclusion with the reduced docking following stimulation and the failures observed in Figure 2aii during later parts of the train. Is the RRP smaller initially, or does it only fail to refill properly? Given the very rapid decreases from the initial response in SYT7 mutants, it seems likely it is smaller already? If not, what leads to a normal RRP that occurs in the absence of STY7 before the stimulation train. Does the system eventually return to the baseline state in the absence of SYT7, but it just takes longer? If so, it seems SYT7 maybe facilitates the rate of these processes, but is clearly not absolutely required for them?

3. Are the authors surprised to see no differences in docked vesicles in the 1 versus 50 stimulation events at 20 Hz at the 5 msec time point of the EM? It seems hard to imagine that 49 more actions potentials at these release sites would not deplete docked vesicles more. Does the author imagine synaptic vesicle replenishment and redocking at 50x occurring in only 5 msec? Hard to believe. Seems more likely something is wrong with the stimulation and they are not really triggering 50 actions potentials due to some technical issue or that the number of vesicles available for release is depleted right away. This measurement seems off.

4. The last half of the described experiments are hard to interpret. These deal with localization and cleavage of SYT7 using virally expressed tagged fusion proteins. It looks like these overexpression approaches result in the targeting of SYT7 everywhere. As can be seen in Figure 4a, the tagged protein seems to go everywhere and fills the neuron. Even the nuclear envelope of the HEK293 cells has SYT7 on it (Supplemental Figure 3f). The authors make claims about co-localization with lysosomal tagged proteins and with the plasma membrane throughout the 2nd half of the work, but I can't figure out how the authors separate these two locales. Where in the manuscript is a co-stain with a plasma membrane marker to show localization to this compartment? The major co-localization they show is with a tagged LAMP1. In order to provide co-localization that would convince a reader the overexpressed SYT7 protein is simply not going everywhere membrane compartments are in the cell is to do a more careful set of co-stains with other compartmental markers and show high resolution images to demonstrate SYT7 does not co-localize with other membrane compartments. For example, in Figure 4d that shows an optical slice of the axon, SYT7 is everywhere – there isn't enough resolution to say much of anything other than its less intense (barely) in the center of the axon. At that level of resolution, it would likely co-label with any marker undergoing axonal transport.

5. Without careful co-localization studies at higher resolution with other membrane compartmental markers (Golgi, ER, SVs, endosomes, plasma membrane), the remaining figures are equally difficult to interpret. Clearly tagged SYT7 is shifting molecular weights a bit on addition of secretase blockers, and its staining intensity changes dramatically with these and palmitoylation manipulations, so something interesting could be happening here. However, I can't interpret the rescue experiments attempting to re-localize SYT7 without better co-localization studies and some quantification of the amount of signal in these compartments compared to the control situation. Are the levels of SYT7 in mistargeted plasma membrane locales 100-fold more than what would normally be found there? It's unknown and could dramatically alter the interpretations of where the protein functions from. Similarly, the fact that it can be on lysosomes and on the plasma membrane for rescuing its function is not really addressed. How do the authors imagine it works from either of these compartments? Overall, I have real difficulty in interpreting the last four figures of the paper without a substantial reworking of the co-localization experiments and better quantification with a broad panel of compartmental markers to convincingly show localization of the protein.

*Reviewer #2:*

This study makes valuable contributions to clearing up significant confusion in the neuroscience literature about the role of Syt7 in the brain. Using sophisticated, state-of-the art technology the authors generate results that contribute significantly to a more concrete picture of how Syt7 deficiency impacts different forms of synaptic vesicle release and synaptic plasticity. In some cases the results are confirmatory of previous reports, but the study fills several gaps in knowledge that advance the field and should be of interest to investigators working on this problem.

The authors also do a nice job of finally showing the subcellular localization of Syt7 in neurons. In view of results reported by other groups, it is not surprising that they could not detect endogenous Syt7 with antibodies in hippocampal neurons. Their observation of tagged Syt7 on lysosomes is consistent with previous reports in other cell types, but the potential role of the lysosomal Syt7 pool in generating the PM pool is not sufficiently explored.

In this reviewer's view, to explore better the neuronal lysosomal population is important for three reasons: (1) the authors found that at least some of the functions requiring Syt7 can be rescued by a construct exclusively targeted to lysosomes. Why? (2) The authors find a role for palmitoylation in the correct targeting/stability of Syt7, and previous work showed that Syt7 traffics to lysosomes by forming a palmitoylation-dependent complex with the lysosomal tetraspanin CD63. From the data presented, it is not clear how the proposed γ-secretase cleavage relates to the need for palmitoylation, and to the functional role of Syt7 in distinct neuronal locations (what do the authors mean by subsynaptic membrane trafficking?). (3) Thin layer chromatography, lipid droplet staining and an indirect method, SYP/SYB interactions, were used to rule out an impact of Syt7 deficiency on the abundance/distribution of cholesterol in neurons. However, the results shown in in Figure S4 are not robust enough to allow this conclusion. Abnormal cholesterol trafficking would indeed represent a direct link between lysosomes and functional abnormalities at the level of the PM, and the results presented do not eliminate this possibility.

Considering the large number of studies that have focused on the role of Syt7 in the brain, there is one important piece of information missing in discussions about this protein: Is there a neurological phenotype in Syt7 KO animals? Have any behavioral abnormalities been reported that would be consistent with what is now known about Syt7 function in the brain? Clear phenotypes have been reported as a consequence of defective insulin secretion and lysosomal exocytosis in Syt7 KO mice, and these phenotypes are all related to a vesicular, not PM localization of Syt7. Unlike what is stated in this paper, there are several commonalities regarding the localization of Syt7 in different tissues – particularly considering the functional similarity between lysosomes and various secretory vesicles. A discussion of these issues would be an important addition, considering the broad readership of *eLife*.

The potential role of the lysosomal Syt7 pool in generating the PM pool is not sufficiently explored. The lysosomal targeting machinery is known to be saturated easily, with excess (overexpressed) protein trafficking instead by default to the PM. This could be the explanation, by the way, to the statement in the discussion that 'small amounts of Lamp1 can be detected on the PM' – this is not the case for endogenous Lamp1, as extensively demonstrated in several mammalian cell types (there is so little Lamp1 on the PM that this protein is a very useful tool to monitor induced lysosomal exocytosis). The authors mention choosing conditions to 'sparsely' express tagged Syt7 – how were these optimal expression levels determined? To rule out PM localization as a consequence of overexpression, it is important to show if different levels of expression of the tagged construct affect the ratio of lysosome vs PM localization.

The images shown for HEK293T cells expressing tagged Lamp1 and Syt7 are puzzling – why so few lysosomes? Were optical sections selected to emphasize the PM localization of Syt7? In non-neuronal cells Syt7 is predominantly targeted to lysosomes, with some detection on the PM that has been interpreted as saturation of the targeting machinery when using overexpressed constructs. Full cellular projections combining all optical sections should be assembled to allow a better idea of where most of the expressed Syt7 is – on lysosomes or the PM. This may be harder to quantify on neurons given their morphology, but in HEK293 cells it should be straightforward and informative.

In this reviewer's view, to explore better the neuronal lysosomal population is important for three reasons: (1) the authors found that at least some of the functions requiring Syt7 can be rescued by a construct exclusively targeted to lysosomes. Why? (2) The authors find a role for palmitoylation in the correct targeting/stability of Syt7, and previous work showed that Syt7 traffics to lysosomes by forming a palmitoylation-dependent complex with the lysosomal tetraspanin CD63. From the data presented, it is not clear how the proposed γ-secretase cleavage relates to the need for palmitoylation, and to the functional role of Syt7 in distinct neuronal locations (what do the authors mean by subsynaptic membrane trafficking?). (3) Thin layer chromatography, lipid droplet staining and an indirect method, SYP/SYB interactions, were used to rule out an impact of Syt7 deficiency on the abundance/distribution of cholesterol in neurons. However, the results shown in in Figure S4 are not robust enough to allow this conclusion. Abnormal cholesterol trafficking would indeed represent a direct link between lysosomes and functional abnormalities at the level of the PM, and the results presented do not eliminate this possibility.

Is there a neurological phenotype in the Syt7 KO animals? Have any behavioral abnormalities been reported that would be consistent with what is now known about Syt7 function in the brain? Clear phenotypes have been reported as a consequence of defective insulin secretion and lysosomal exocytosis in Syt7 KO mice, and these phenotypes are all related to a vesicular, not PM localization of Syt7. Unlike what is stated in this paper, there are several commonalities regarding the localization of Syt7 in different tissues – particularly considering the functional similarity between lysosomes and various secretory vesicles. A discussion of these issues would be an important addition, considering the broad readership of *eLife*.

[Editors’ note: further revisions were suggested prior to acceptance, as described below.]

Thank you for submitting your article "Synaptotagmin 7 is targeted to the axonal plasma membrane through γ-secretase processing to promote synaptic vesicle docking in mouse hippocampal neurons" for consideration by *eLife*. Your article has been reviewed by 2 peer reviewers, and the evaluation has been overseen by a Reviewing Editor and Kenton Swartz as the Senior Editor. The following individual involved in review of your submission has agreed to reveal their identity: Haoxing Xu (Reviewer #3).

The reviewers have discussed their reviews with one another, and the Reviewing Editor has drafted this letter to help you prepare a revised submission.

Please address the comments of reviewer 3 using textual changes in the manuscript and write a rebuttal.

*Reviewer #1:*

The revised manuscript by Vevea et al. adds some new experimental data and most importantly, tones down a lot of the claims in the 2nd half of the manuscript on Syt7 processing by presenilin and the role of subcellular localization of the protein. I think figures 1-4 make a nice contribution to the Syt7 mutant analysis by using GluSnFR and flash-freeze EM to catalog a number of defects previously reported separately by multiple groups. I still have difficulty interpreting figures 5-8 that deal with Syt7 localization, even though the authors have added new data using a method for tagging endogenous Syt7 to their analysis. The resolution is just to low for me to conclude Syt7 is only on the plasma membrane and lysosomes at synaptic terminals. I will leave determination of the quality of the localization data to the other reviewer, as I don't feel comfortable reviewing these panels. A simple addition of a gradient fractionation experiment from synaptosome preps to show most Syt7 localizes with other plasma membrane proteins would really help this reviewer feel more comfortable with their localization claims. However, the rescue experiments with mistargeted constructs does provide evidence Syt7 directed to SVs cannot rescue release, while tethering it to the PM can, so that data seems fine to me. Overall, I still have difficulty interpreting parts of the last half of the paper, but the authors have significantly toned down their prior claims, which I appreciate.

*Reviewer #3:*

In this resubmitted manuscript by Vevea at al., the authors reported the roles of Syt-VII, a ca^2+^ effector associated with various membranes, in multiple steps of synaptic vesicle exocytosis in cultured hippocampal neurons. Using optical GluSnFR imaging and Zap-freeze EM, they showed that Syt-VII regulates activity-dependent synaptic vesicle (SV) docking and replenishment. These results are mostly convincing and support previous conclusions that were drawn based on the electrophysiological measurements. In addition, Syt-VII are shown to be localized to both axonal and lysosomal membranes via γ-secretase processing and palmitoyltion. As Syt-VII is localized to the axonal membrane, SV membranes, and lysosomal membranes, three types of targeting constructs were used to rescue the synaptic defects in Syt-VII KO neurons. Overall, the authors responded constructively to the reviewers' comments. A number of additional control experiments were performed during revision. For example, using CRIPSR/cas9 to knock-in a tag, the localization of Syt-VII is now confirmed in the endogenous setting. The only concern that I have is related to the function of lysosomal vs. axonal localization of Syt-VII. It is not clear whether lysosome-localized Syt-VII may indeed contribute to SV cycle. Given that neuronal firing can induce ca^2+^ release from lysosomes (Pada,sey et al., Neuron 2017; PMID 27989455), it is possible that lysosomal Syt-VII may traffic to the axonal membrane via lysosomal exocytosis. I understand that the Lamp1-Syt-VII rescue experiment may not be as clean as expected. Are there other ways to separate the effects of these two pools of Syt-VII? For example, palmitoyltion of PM-localized Syt-VII may be mediated by acyl-protein thioesterases in the cytosol while palmitoyltion of lysosome-localized Syt-VII may be catalyzed by palmitoyl protein thioesterases in lysosomes. I am assuming that 2-BP is a general inhibitor of palmitoylation. Are there any specific inhibitors available for cytosolic vs. lysosomal thioesterases? On the other hand, since lysosomal CD63 protein is required for Syt-VII targeting to lysosomes, it might be more informative if the lysosome-targeted rescue construct is based on CD63 (instead of Lamp1), with its tyrosine targeting motif mutant as the negative control. Are there any synaptic defects, e.g., in PPT and asynchronous release, in CD63 KO neurons? Although I am not one of the original reviewers of this revised manuscript, as the manuscript is already quite comprehensive, I will let the reviewer editor decide whether it is essential for the authors to address these comments by performing new experiments.

---

## [Author Response]

[Editors’ note: The authors appealed the original decision. What follows is the authors’ response to the first round of review.]

Reviewer #1:Vevea et al. describe an analysis of Synaptotagmin 7 function in neurotransmission in cultured hippocampal neurons. The authors re-capitulate several previously published phenotypes that were found via electrophysiological approaches with optical GluSnFR analysis in the current study. They also employ zap and freeze EM to identify a small defect in SV docking and SV replenishment after high frequency stimulation. The authors then turn to examine SYT7 localization and suggest a role for γ-secretase cleavage of the protein in facilitating its axonal localization. They argue for SYT7 co-localizing with both lysosomes and the plasma membrane, but most of these experimental approaches are quite confusing and are hard to interpret without more careful localization analysis with a broader panel of compartmental markers. Finally, the author use a mistargeting strategy to suggest SYT7 can reside on either lysosomes or the plasma membrane to facilitate the majority of its functional roles in SV trafficking. Again, I find these particular aspects of the study difficult to interpret, as the localization methods do not appear to be very clean and they are strongly confounded by protein levels – one has no idea of the relative amount of SYT7 compared to the normal amounts in each locale, and reduced or enhanced levels could severely confound the strong interpretations the authors have made. Overall, although the 1st half of the work validates some prior observations in the field and extends the work with EM analysis, the second half is quite challenging to interpret and place in an appropriate context.1. The first several figures of the paper generally recapitulate work from the field, including the authors' own electrophysiological analysis (Liu et al., 2014 eLife), on SYT7 knockout mouse lines. The authors find reduced paired-pulse facilitation and a reduction in synaptic vesicle replenishment rate during high frequency stimulation. They also note a very small change in asynchronous release (~3%) in the absence of SYT7. The main advance here is the use of the GluR sniffer approach versus electrophysiology. The drawback, however, is the variability in the optical method, as it depends on the number of release sites imaged per ROI in each culture. The authors did not quantify single synaptic vesicle signals (minis), so the quantal response from GluR sniffer δ f signals at each site seems unknown. Whether 1 or 50 synaptic vesicles are released during each stimulation is unclear without a baseline quantal signal. Therefore, it seems difficult to make any quantitative comments about single stimulation responses across genotypes or even across ROIs within the same genotype. Only changes from the initial response of an ROI in each culture seem robustly quantifiable in terms of an absolute value. This variability is obvious in Figure 2a, where responses to individual stimuli both within the same field of view or different field of views (A-E) bounce around significantly. As such, the value of the imaging approach in cultured neurons over more well-characterized electrophysiological approaches in defined neuronal populations from slices is unclear.

We argue that there is a lot of utility in attempting to measure all of the disparate observed synaptic phenotypes at the same ‘workhorse neurons’ that are so widely used in the field: dissociated hippocampal neurons. Furthermore, we use a new optical method to directly detect glutamate release, iGluSnFR. The reviewer notes variability in the method; we agree that there is some variability (as in most biological related measurements, including EPSCs), but the data are very clear and reproducible from trial to trial. We find this approach quite robust. We note that data generated via somatic voltage clamp electrophysiology, which is the standard method to monitor synaptic transmission, are prone to measurement error and artifacts from incomplete voltage clamping of dendritic spines (due to cable filtering; (Williams, S.R. and Mitchell, S.J. (2008) *Nat. Neuro.*)). Nevertheless, we appreciate the reviewer’s critique and applied their suggestion to perform quantal analysis on our dataset.

To define a single quantum, we make the assumption that – late in high frequency stimulus (HFS) trains – single SV release predominates, aka single quanta. Therefore, we took our release data from late in the train, binned it in a histogram and found a Gaussian distribution centered around 0.14 DF/F_o_ iGluSnFR units (Figure 2h). We defined this as a single quantum and used this value to interpret initial release from WT neurons (finding it multiquantal, Figure 2g) and asynchronous release from WT and SYT7KO neurons (finding uni- and multiquantal asynchronous release, to our surprise (Figure 2i), and decreased multiquantal asynchronous release from SYT7KO neurons (Figure 2j)). We realize that multiquantal release at hippocampal synapses is a debated issue, but our data strongly support the conclusion that this does occur. Additionally, recent electron microscopy studies provide strong, independent evidence that multiquantal release is common in this preparation (Kusick G., et al. (2020) *Nat. Neuro*).

We thank the reviewer for this insightful suggestion. We feel that this analysis has improved the clarity of our study and bolstered confidence of our protocol and analysis. This analysis further validates the utility of iGluSnFR imaging to characterize synaptic function. Indeed, describing multiquantal asynchronous release, and showing that this involves SYT7, is an important new conclusion for the field.

2. The authors previously reported the RRP was not altered using electrophysiology measurements, but it appears their optical and EM approach suggest a different conclusion with the reduced docking following stimulation and the failures observed in Figure 2aii during later parts of the train. Is the RRP smaller initially, or does it only fail to refill properly? Given the very rapid decreases from the initial response in SYT7 mutants, it seems likely it is smaller already? If not, what leads to a normal RRP that occurs in the absence of STY7 before the stimulation train. Does the system eventually return to the baseline state in the absence of SYT7, but it just takes longer? If so, it seems SYT7 maybe facilitates the rate of these processes, but is clearly not absolutely required for them?

We argue that the EM analysis does not suggest an RRP defect, only a defect in closely docked vesicles. The RRP likely encompasses the entire docked and tethered vesicle pool (Maus L., et al. (2020) *Cell Rep.*), which is an important distinction. Indeed, in our experiments, the vesicle pool near the active zone (i.e. within 100 nm) does not appear to be changed (Figure 3e). We only observed differences in the docked pool, 0-5 nm from the active zone membrane, and no differences in the SV pool 5-100 nm from the active zone. Without the rapid ‘zap-and freeze’ approach, the change in closely docked vesicles would likely have been overlooked.

Our data (Figure 3c) show that the docked pool is the same prior to stimulation; only after stimulation is there a defect in the number of docked vesicles. We agree with the reviewer’s suggestion, this observation is consistent with the role of SYT7 as a protein that facilitates activity-dependent docking of SVs. Given enough time after stimulation, docked vesicles return to baseline. So yes, we interpret our data, and the data of others, to mean that SYT7 facilitates release during activity; this manifests as facilitation during PPR experiments, increased asynchronous during HFS trains, and resistance to depression. Our view is that SYT7 plays a positive role in the dynamic, activity dependent, docking reaction, thereby enhancing release under some conditions.

3. Are the authors surprised to see no differences in docked vesicles in the 1 versus 50 stimulation events at 20 Hz at the 5 msec time point of the EM? It seems hard to imagine that 49 more actions potentials at these release sites would not deplete docked vesicles more. Do the author imagine synaptic vesicle replenishment and redocking at 50x occurring in only 5 msec? Hard to believe. Seems more likely something is wrong with the stimulation and they are not really triggering 50 actions potentials due to some technical issue or that the number of vesicles available for release is depleted right away. This measurement seems off.

Synaptic replenishment is very fast. A major mystery in the field has been: how can SV replenishment occur so quickly (because vesicles are not completely depleted during long HFS trains), while known SV endocytosis rates occur so slowly (seconds to tens of seconds)? Work by Tim Ryan and Erik Jorgensen/Shigeki Watanabe demonstrated a form of ultrafast endocytosis that occurs on the tens to hundreds of ms timescale. Now Shigeki Watanabe is describing a form of activity dependent ultrafast transient undocking/docking reactions that occur on the ms timescale after an action potential (Kusick G., et al. (2020) *Nat. Neuro.*). Prior to these recent developments, we agree that our results may have been surprising. However, considering these recent reports, our results and interpretations are sensible. Current experiments suggest the docked pool is a fraction of the total RRP and the processes controlling RRP replenishment and docking may be different processes. Based on our experiments, and from work by other groups, including Christian Rosenmund’s lab, we argue that a mechanism involving SYT1 and SYT7 leads to transient, activity dependent, vesicle docking. Our view is that SYT7 promotes transient docking, but it is clearly not absolutely needed for transient docking to occur (Figure 3c). Furthermore, as to the reliability of the stimulus given for the EM experiments, many lines of evidence support its robustness. Single stimulus and HFS trains were monitored and verified by a voltmeter and monitored by cellular uptake of FM1-43 dye (Kusick G., et al., (2020) *Nat. Neuro.*). Additionally, the observation of pits in the active zone 5 ms after the 50 AP HFS train strongly supports the conclusion that stimulation continued to the end of the train.

4. The last half of the described experiments are hard to interpret. These deal with localization and cleavage of SYT7 using virally expressed tagged fusion proteins. It looks like these overexpression approaches result in the targeting of SYT7 everywhere. As can be seen in Figure 4a, the tagged protein seems to go everywhere and fills the neuron. Even the nuclear envelope of the HEK293 cells has SYT7 on it (Supplemental Figure 3f). The authors make claims about co-localization with lysosomal tagged proteins and with the plasma membrane throughout the 2nd half of the work, but I can't figure out how the authors separate these two locales. Where in the manuscript is a co-stain with a plasma membrane marker to show localization to this compartment? The major co-localization they show is with a tagged LAMP1. In order to provide co-localization that would convince a reader the overexpressed SYT7 protein is simply not going everywhere membrane compartments are in the cell is to do a more careful set of co-stains with other compartmental markers and show high resolution images to demonstrate SYT7 does not co-localize with other membrane compartments. For example, in Figure 4d that shows an optical slice of the axon, SYT7 is everywhere – there isn't enough resolution to say much of anything other than its less intense (barely) in the center of the axon. At that level of resolution, it would likely co-label with any marker undergoing axonal transport.

We have added experiments localizing endogenously expressed SYT7 via CRISPR/HITI (i.e. we knocked-in a tag) and have added further clarification in the text regarding how we interpret these experiments. Importantly, we must clarify that while we first identified cleavage of SYT7 through overexpressed protein, the main figure (Figure 5a-c) is focused on the endogenous protein. Again, we first observed cleavage of overexpressed protein, then we confirmed this observation and performed an IC50 curve with native protein.

We also respectfully disagree with this reviewer that the overexpressed protein results in the targeting of SYT7 everywhere. In Figure 4a, SYT7 protein is asymmetrically enriched to axons, and is largely absent from dendrites. So, gently over-expressed SYT7 does not appear to non-specifically spilling over into the plasma membrane; rather, it is targeted to axons. The reviewer asked where we included a co-stain of a plasma membrane marker in our study, and we note that this was included in the original manuscript in Figure 4 —figure supplement 1a. We showed axonal enrichment by transiently transfecting a general PM-msGFP marker and a SYT7-HaloTag construct using a bicistronic P2A vector. We have now included a line scan demonstrating enrichment of SYT7 to the axon versus dendrites. PM-msGFP is targeted to all plasma membranes, everywhere and evenly, while SYT7 is – again – enriched in axons vs dendrites. This transfection method (which makes it more difficult to control the expression of constructs) results in some spillover of SYT7 to dendritic plasma membrane (due to higher levels of over-expression), but SYT7 is still enriched in the axon. Using sparse lentiviral transduction (which is a better method to control protein levels vs transient transfection), exogenously expressed SYT7 was mainly localized to the axon, with some signal also in the soma; we show that these latter structures are predominantly LAMP1 positive membranes.

We removed the HEK293 image examples from this supplement because it did not add new information to the manuscript (we note that it is not surprising that a secretory protein, when overexpressed in HEK cells, localizes to the nuclear membrane as this membrane is continuous with the rough ER where membrane proteins are first translated).

Most importantly however, we have localized endogenously expressed, carboxy tagged SYT7 using CRISPR/HITI, as described in Suzuki K., et al., (2016) *Nature* and Willems J., et al. (2020) *PLoS Biol*. These experiments confirmed our previous localization results. In these HITI knock-in experiments, the signal from SYT7 is very weak (expected due to low expression of endogenous SYT7), but by using HILO microscopy and Airyscan imaging with ICC amplification, we can state that endogenous SYT7 is asymmetrically localized to the axon and present in the soma, consistent with our previous localization experiments. We thank the reviewer for encouraging us to perform this type of experiment, as it strengthened our original conclusions.

In the original version of this manuscript, we co-stained for a variety of compartment markers but, for space reasons, we did not show example images of these results; instead chose to summarize the quantified localization data in Figure 6d. We agree with the reviewer that our original presentation of these data does not give the reader an understanding of the degree of co-localization of SYT7 with these markers on an absolute basis. Therefore, in the revised manuscript (and because *eLife* allows multiple supplements for each figure), we have included sample super-resolution images and absolute colocalization data from the control condition from Figure 6d, in Figure 4 —figure supplement 2f-g. Examining the soma, we confirm co-localization with LAMP1 with two different protocols, and we co-stain with markers for ER, *cis*-Golgi, *trans*-Golgi, M6PR and sortilin positive post Golgi vesicles, and early endosomes. In addition to lysosomes, we observed a smaller amount of colocalization with the *trans*-Golgi and sortilin vesicles.

For Figure 4d and Figure 4 —figure supplement 1c-d, our argument is that SYT7 is present, not on SV clusters, but rather on the plasma membrane. We agree with the reviewer when they state that the ultimate future goal is to elucidate the exact location of SYT7 not only at steady state, with respect to SVs, but also during activity; however, that is a major undertaking that will involve multiple advanced techniques to make it possible to detect low copy numbers of SYT7 that we suspect may play a role in aspects of endosomal trafficking in nerve terminals. So, this aspect of the work will require several more years.

5. Without careful co-localization studies at higher resolution with other membrane compartmental markers (Golgi, ER, SVs, endosomes, plasma membrane), the remaining figures are equally difficult to interpret. Clearly tagged SYT7 is shifting molecular weights a bit on addition of secretase blockers, and its staining intensity changes dramatically with these and palmitoylation manipulations, so something interesting could be happening here. However, I can't interpret the rescue experiments attempting to re-localize SYT7 without better co-localization studies and some quantification of the amount of signal in these compartments compared to the control situation. Are the levels of SYT7 in mistargeted plasma membrane locales 100-fold more than what would normally be found there? It's unknown and could dramatically alter the interpretations of where the protein functions from. Similarly, the fact that it can be on lysosomes and on the plasma membrane for rescuing its function is not really addressed. How do the authors imagine it works from either of these compartments? Overall, I have real difficulty in interpreting the last four figures of the paper without a substantial reworking of the co-localization experiments and better quantification with a broad panel of compartmental markers to convincingly show localization of the protein.

We have included additional co-localization data and analysis, which includes all these requested compartments, and more, as detailed above. Also, as mentioned above, in the original manuscript we plotted the changes to Pearson’s r between control and +DAPT conditions (Figure 6d).

Regarding the retargeting rescue SYT7 experiments, because endogenous SYT7 is expressed at such a low level, single transduction events with rescue constructs (including WT, unmodified SYT7) result in overexpression. Importantly though, for the WT SYT7 rescue, all phenotypes are rescued without any additional ‘off-target/gain of function’ effects. Our retargeted construct expression levels were carefully tittered to match the same level of expression of the WT protein that rescued synaptic function in the KO. Our conclusion is that a very small amount of SYT7 is needed to carry out SYT7 function – supporting vesicle docking during activity – but more SYT7 protein (to a point, note recent work by Fujii, T., et al., (2021) *Sci. Reports*) does not result in some sort of gain-of-function, as we do not see evidence for that in any of our experiments.

Finally, we agree that the data obtained with the lysosome-SYT7 rescue construct is difficult to interpret. However, we decided to include these data for the sake of transparency and completeness. Because we observed a docking defect in the KO, and because the PM-SYT7 construct completely rescues all phenotypes associated with the KO, we propose that SYT7 functions at the PM to support vesicle docking. How does lysosomal SYT7 influences the SV cycle? This is difficult to say but may be explained by transient trafficking of this construct through the axonal PM, as explained in the revised text. Although we do not detect this construct on the axonal PM upon rescue-level expression, upon massive overexpression with lipid-mediated transfection reagents, we can ‘push’ this construct to what appears to be the axonal PM (Figure 7 —figure supplement 1b), suggesting that it may be there at undetectable levels at normal rescue expression levels. Interestingly, if this is true, it still tells us something about SYT7 as this construct rescued PPF and depression, but not train related asynchronous release, therefore train related asynchronous release may require higher levels of SYT7.

Reviewer #2:This study makes valuable contributions to clearing up significant confusion in the neuroscience literature about the role of Syt7 in the brain. Using sophisticated, state-of-the art technology the authors generate results that contribute significantly to a more concrete picture of how Syt7 deficiency impacts different forms of synaptic vesicle release and synaptic plasticity. In some cases the results are confirmatory of previous reports, but the study fills several gaps in knowledge that advance the field and should be of interest to investigators working on this problem.The authors also do a nice job of finally showing the subcellular localization of Syt7 in neurons. In view of results reported by other groups, it is not surprising that they could not detect endogenous Syt7 with antibodies in hippocampal neurons. Their observation of tagged Syt7 on lysosomes is consistent with previous reports in other cell types, but the potential role of the lysosomal Syt7 pool in generating the PM pool is not sufficiently explored.

We thank the reviewer for their critical feedback. We have added experiments to the manuscript to examine, in more detail, the subcellular localization of SYT7 in the soma to, in turn, help identify the secretory trafficking pathway that SYT7 uses to reach its destination(s) (Figure 4 —figure supplement 2f-g). We have also conducted experiments using CRISPR/HITI to tag the genomic locus of SYT7 and assess localization of endogenously expressed protein (Figure 3 —figure supplement 1e-g). We interpret these HITI experiments as confirming the axonal targeting and asymmetric distribution of SYT7 in neurons. We modified some of our discussion, based on this reviewer’s input, concerning SYT7 localization in other cell types and potential behavioral or coordination related phenotypes in the whole animal. Again, this review has been particularly helpful in revising our manuscript.

The potential role of the lysosomal Syt7 pool in generating the PM pool is not sufficiently explored. The lysosomal targeting machinery is known to be saturated easily, with excess (overexpressed) protein trafficking instead by default to the PM. This could be the explanation, by the way, to the statement in the discussion that 'small amounts of Lamp1 can be detected on the PM' – this is not the case for endogenous Lamp1, as extensively demonstrated in several mammalian cell types (there is so little Lamp1 on the PM that this protein is a very useful tool to monitor induced lysosomal exocytosis). The authors mention choosing conditions to 'sparsely' express tagged Syt7 – how were these optimal expression levels determined? To rule out PM localization as a consequence of overexpression, it is important to show if different levels of expression of the tagged construct affect the ratio of lysosome vs PM localization.

These are excellent points, and we are happy to clarify these valid concerns. First, all constructs are expressed at the same levels, and that level was what was needed to rescue the KO phenotypes with WT SYT7; a representative immunoblot, showing the expression level, is included in Figure 7 —figure supplement 1g. All constructs are expressed at a higher level than endogenous SYT7 however, and this is due to the low level of endogenous expression. Checking our viral titers with ICC, we see that going lower than these expression levels results in less than 100% of neurons being transduced. Therefore, the expression level observed for rescue is the result of every cell being transduced approximately once and the relative strength of the hSynapsin promoter versus the native SYT7 promoter elements.

Importantly, we made progress localizing endogenously expressed SYT7 by using CRISPR/HITI protocols (Suzuki et al. *Nature* 2016 and Willems et al. *PLoS Biol.* 2020) to tag the endogenous locus of SYT7 with HaloTag. Although it was difficult to detect, due to the low expression of native SYT7 in dissociated neurons, we were able to use HILO microscopy to localize (JF549 signal) the tagged construct to axons and a compartment in the soma that is consistent with lysosome morphology (Figure 4 —figure supplement 1e). Additionally, we attempted to boost the signal through ICC and an anti HaloTag antibody, and while this was partially successful (axons stained brightly and showed clear asymmetry to other neurites) (Figure 4 —figure supplement 1f), the soma stained non-specifically (all soma contained signal) and so lysosome localization was difficult to confirm. Considering the totality of our experiments, and the total size of an axon, we estimate more than half of SYT7 resides on the axon plasma membrane.

The images shown for HEK293T cells expressing tagged Lamp1 and Syt7 are puzzling – why so few lysosomes? Were optical sections selected to emphasize the PM localization of Syt7? In non-neuronal cells Syt7 is predominantly targeted to lysosomes, with some detection on the PM that has been interpreted as saturation of the targeting machinery when using overexpressed constructs. Full cellular projections combining all optical sections should be assembled to allow a better idea of where most of the expressed Syt7 is – on lysosomes or the PM. This may be harder to quantify on neurons given their morphology, but in HEK293 cells it should be straightforward and informative.

The reviewer is correct, we did use an optical section to visualize plasma and lysosomal membrane localization. However, since it is not integral to our study and it has caused confusion during the review process, we elected to remove it. It is difficult to interpret the HEK293T data as anything other than rough localization to PM and lysosome membranes, which was the only point we intended to make.

In this reviewer's view, to explore better the neuronal lysosomal population is important for three reasons: (1) the authors found that at least some of the functions requiring Syt7 can be rescued by a construct exclusively targeted to lysosomes. Why? (2) The authors find a role for palmitoylation in the correct targeting/stability of Syt7, and previous work showed that Syt7 traffics to lysosomes by forming a palmitoylation-dependent complex with the lysosomal tetraspanin CD63. From the data presented, it is not clear how the proposed γ-secretase cleavage relates to the need for palmitoylation, and to the functional role of Syt7 in distinct neuronal locations (what do the authors mean by subsynaptic membrane trafficking?). (3) Thin layer chromatography, lipid droplet staining and an indirect method, SYP/SYB interactions, were used to rule out an impact of Syt7 deficiency on the abundance/distribution of cholesterol in neurons. However, the results shown in in Figure S4 are not robust enough to allow this conclusion. Abnormal cholesterol trafficking would indeed represent a direct link between lysosomes and functional abnormalities at the level of the PM, and the results presented do not eliminate this possibility.

We thank the reviewer for their detailed points and insightful questions, and we agree that the original text was not as clear as it could have been. We have clarified these issues, as detailed below.

Lysosome rescue:

The reviewer makes a great point and touches on a piece of data that we had difficulty interpreting. With the reviewer’s feedback and additional internal discussions, we have toned-down our interpretation of the lysosome-targeted rescue construct as we cannot rule out excess protein trafficking to the PM. We note that we do not detect the LAMP1-S7 construct in axons under rescue level conditions (Figure 7 —figure supplement 1g), but that during transfection with high amounts of DNA, we sometimes did detect this construct in axons (Figure 7 —figure supplement 1b). However, it is interesting to note that asynchronous release was not rescued with this construct. If a small amount of this construct localizes to the PM, then we could hypothesize that a small amount of SYT7 is needed to support PPF and resistance to depression but is not sufficient to drive asynchronous release. We prefer to leave the data in the manuscript for transparency and completeness but have added caveats in the Results and Discussion sections of our revised study. If the data regarding lysosome targeted SYT7 are felt to be a distraction, we can of course remove these data entirely.

Post-translational modifications:

We observed that endogenous SYT7 is cleaved by the intramembrane protease g-secretase; cleavage of proteins by g-secretase normally releases them from the membrane. We observe SYT7 is still associated with membranes (lysosome and axon PM) but is sensitive to treatment with 2-BP, a palmitoylation inhibitor. We therefore argue that SYT7 becomes a peripheral membrane protein that is dependent on palmitoylation to associate with membranes after cleavage by g-secretase. We hypothesize that rapid depalmitoylation/palmitoylation cycles might allow SYT7 to rapidly transfer between different membranes during the SV cycle (subsynaptic membrane trafficking), where membranes are constantly endocytosed and repackaged into SVs. This is currently the subject of a follow-up study that addresses a second possible function of SYT7 in the resolution of endosomes before returning to the PM where most of the protein resides at steady state.

SYT7/cholesterol metabolism theory:

We conducted several experiments to assess cholesterol trafficking in SYT7KO neurons. As the reviewer notes, this included TLC of total lipids, direct localization of neutral lipids, and SYP/SYB cholesterol dependent interactions. TLC measures bulk lipids and is a crude measure of total lipids, but directly measuring neutral lipids like cholesterol with a solvochromatic dye, and then probing cholesterol dependent interactions of proteins that rely on PM cholesterol is quite sensitive (SYP/SYB cholesterol dependent interaction). Indeed, in NPC mutants (Liscum L., et al., (1989) *J. Cell. Biol.*) cholesterol accumulates in lysosomes. We note that Chu B., et al., ((2015) *Cell*) reported that silencing SYT7 or expressing a dominant negative form of SYT7 increases the amount of cholesterol build-up in LAMP1+ structures (Figure 4c and h from Chu B., et al.). However, in our experiments, we do not observe a role for SYT7 in the trafficking of cholesterol. Indeed, recent screens have failed to reproduce the links between SYT7-peroxisome-cholesterol metabolism pathway that Chu B., et al. described (Trinh M.N., et al., (2020) *PNAS*).

Is there a neurological phenotype in the Syt7 KO animals? Have any behavioral abnormalities been reported that would be consistent with what is now known about Syt7 function in the brain? Clear phenotypes have been reported as a consequence of defective insulin secretion and lysosomal exocytosis in Syt7 KO mice, and these phenotypes are all related to a vesicular, not PM localization of Syt7. Unlike what is stated in this paper, there are several commonalities regarding the localization of Syt7 in different tissues – particularly considering the functional similarity between lysosomes and various secretory vesicles. A discussion of these issues would be an important addition, considering the broad readership of eLife.

This is a good point and we have included discussion concerning the localization of SYT7, and phenotypes related to SYT7 vesicular localization, based on the reviewer’s comments. We have also added a short description of reported behavioral abnormalities in SYT7 KO mice.

[Editors’ note: what follows is the authors’ response to the second round of review.]

Reviewer #1:The revised manuscript by Vevea et al. adds some new experimental data and most importantly, tones down a lot of the claims in the 2nd half of the manuscript on Syt7 processing by presenilin and the role of subcellular localization of the protein. I think figures 1-4 make a nice contribution to the Syt7 mutant analysis by using GluSnFR and flash-freeze EM to catalog a number of defects previously reported separately by multiple groups. I still have difficulty interpreting figures 5-8 that deal with Syt7 localization, even though the authors have added new data using a method for tagging endogenous Syt7 to their analysis. The resolution is just to low for me to conclude Syt7 is only on the plasma membrane and lysosomes at synaptic terminals. I will leave determination of the quality of the localization data to the other reviewer, as I don't feel comfortable reviewing these panels. A simple addition of a gradient fractionation experiment from synaptosome preps to show most Syt7 localizes with other plasma membrane proteins would really help this reviewer feel more comfortable with their localization claims. However, the rescue experiments with mistargeted constructs does provide evidence Syt7 directed to SVs cannot rescue release, while tethering it to the PM can, so that data seems fine to me. Overall, I still have difficulty interpreting parts of the last half of the paper, but the authors have significantly toned down their prior claims, which I appreciate.

We thank the reviewers for their continued constructive feedback. We agree that the synaptic vesicle vs plasma membrane rescue data support the conclusion that SYT7 executes its function from the axonal plasma membrane, which is consistent with our imaging data (both our new knock-in as well as our mild over-expression data). The congruence of these functional data with the localization data increases confidence in the axonal localization. In addition, we now include references to previous mass spectrometry experiments that have repeatedly and unanimously failed to detect SYT7 on synaptic vesicles (SV). However, mass spec can miss low abundance proteins, so that is why we decided to further test this hypothesis with a rescue experiment using SV-restricted SYT7, finding no rescue of synaptic function. So, SYT7 is undetectable on SVs using mass spectrometry and is non-functional when restricted to the SV membrane. Finally, we note that the fractionation experiment proposed by the reviewer has been performed and the SYT7 signal was absent from the SV fraction but strongly enriched in the ‘synaptic plasma membrane’ fraction (Sugita et al. (2001) *Neuron*). We have included this citation in our revised manuscript as further support for the localization of SYT7 to the plasma membrane.

We have added the following at line 256: **“**Indeed, mass spectrometry analysis of purified SVs fail to identify SYT7 as a principle component (Takamori et al., 2006), and SYT7 has been reported to be enriched in the synaptic plasma membrane in earlier fractionation experiments (Sugita et al., 2001).”

Reviewer #3:In this resubmitted manuscript by Vevea at al., the authors reported the roles of Syt-VII, a ca^2+^ effector associated with various membranes, in multiple steps of synaptic vesicle exocytosis in cultured hippocampal neurons. Using optical GluSnFR imaging and Zap-freeze EM, they showed that Syt-VII regulates activity-dependent synaptic vesicle (SV) docking and replenishment. These results are mostly convincing and support previous conclusions that were drawn based on the electrophysiological measurements. In addition, Syt-VII are shown to be localized to both axonal and lysosomal membranes via γ-secretase processing and palmitoyltion. As Syt-VII is localized to the axonal membrane, SV membranes, and lysosomal membranes, three types of targeting constructs were used to rescue the synaptic defects in Syt-VII KO neurons. Overall, the authors responded constructively to the reviewers' comments. A number of additional control experiments were performed during revision. For example, using CRIPSR/cas9 to knock-in a tag, the localization of Syt-VII is now confirmed in the endogenous setting. The only concern that I have is related to the function of lysosomal vs. axonal localization of Syt-VII. It is not clear whether lysosome-localized Syt-VII may indeed contribute to SV cycle. Given that neuronal firing can induce ca^2+^ release from lysosomes (Pada,sey et al., Neuron 2017; PMID 27989455), it is possible that lysosomal Syt-VII may traffic to the axonal membrane via lysosomal exocytosis. I understand that the Lamp1-Syt-VII rescue experiment may not be as clean as expected. Are there other ways to separate the effects of these two pools of Syt-VII? For example, palmitoyltion of PM-localized Syt-VII may be mediated by acyl-protein thioesterases in the cytosol while palmitoyltion of lysosome-localized Syt-VII may be catalyzed by palmitoyl protein thioesterases in lysosomes. I am assuming that 2-BP is a general inhibitor of palmitoylation. Are there any specific inhibitors available for cytosolic vs. lysosomal thioesterases? On the other hand, since lysosomal CD63 protein is required for Syt-VII targeting to lysosomes, it might be more informative if the lysosome-targeted rescue construct is based on CD63 (instead of Lamp1), with its tyrosine targeting motif mutant as the negative control. Are there any synaptic defects, e.g., in PPT and asynchronous release, in CD63 KO neurons? Although I am not one of the original reviewers of this revised manuscript, as the manuscript is already quite comprehensive, I will let the reviewer editor decide whether it is essential for the authors to address these comments by performing new experiments.

We thank the reviewer for thoroughly reading our manuscript and appreciate their detailed experimental suggestions. The suggested experiments are clever and would be appropriate for a future follow-up study.

We agree with the reviewer that it is still unclear if, or how, lysosomal SYT7 influences the SV cycle. Indeed, our LAMP1-SYT7 rescue experiments were more ambiguous than we had hoped; we currently believe there is an extremely small fraction of LAMP1-SYT7 present on the axonal plasma membrane [undetectable at rescue expression levels (Figure 7b), but detectable with massive over-expression (Figure 7 —figure supplement 1b)].

Our understanding is that 2-BP is a general inhibitor of palmitoylation. Rather than rely on additional inhibitors, we are currently conducting knockdown experiments to identify the enzyme responsible for SYT7 palmitoylation, and this will be a major focus of our next SYT7 study.

To restrict SYT7 to lysosomes we selected a protein (LAMP1) that was primarily localized to these organelles. This is a difficult task because – as the reviewer is aware – there is considerable membrane trafficking between the Golgi, plasma membrane, endosomes, and lysosomes. CD63 is an interesting candidate for building chimeras because it normally interacts with SYT7 on the PM and is required for trafficking to lysosomes (Flannery A.R. et al., JCB 2010). However, from our reading of the literature, the localization of CD63 seemed too unrestrained, e.g., we felt there would be too much CD63 – for our purposes – on the plasma membrane. All of the proteins that we considered for the SYT7 lysosomal restriction experiment are localized to more than just lysosomes and so we made a choice and selected what we thought was the best candidate protein to build our chimeras (i.e., a protein that is mostly localized to lysosomes). We continue to seek better ways to restrict SYT7 to lysosomes for future experiments (this requires a fair bit of protein engineering, which is ongoing). The reviewer also asks if CD63 null synapses display any defects in short-term synaptic plasticity or asynchronous release but there do not seem to be any reports that address this question. This is a great suggestion for a future research project and appreciate the reviewer’s suggestion.

We have added the following at line 530: “These studies reveal the complex regulation of SYT7 by an unknown acyltransferase and the g-secretase complex. Future studies will focus on identifying the enzymes responsible for the rapid palmitoylation and depalmitoylation of SYT7, and the impact of disease-associated presenilin mutations on the processing and trafficking of this protein.”

We have added the following at line 555: **“**When transfected with high amounts of DNA (i.e., when greatly over-expressed), the LAMP1-SYT7a construct was detected on the PM. We note that neuronal activity can induce ca^2+^ release from lysosomes (McGuinness et al., 2007) and trigger lysosomal exocytosis (Padamsey et al., 2017), potentially providing a pathway for the delivery of this chimeric construct to the PM.”